# Graph Neural Networks Exponentially Lose Expressive Power for Node Classification

**Kenta Oono[1, 2], Taiji Suzuki[1, 3]**
{kenta_oono, taiji}@mist.i.u-tokyo.ac.jp
[1]The University of Tokyo [2]Preferred Networks, Inc.
[3]RIKEN Center for Advanced Intelligence Project (AIP)

## Abstract

Graph Neural Networks (graph NNs) are a promising deep learning approach for analyzing graph-structured data. However, it is known that they do not improve (or sometimes worsen) their predictive performance as we pile up many layers and add non-lineality. To tackle this problem, we investigate the expressive power of graph NNs via their asymptotic behaviors as the layer size tends to infinity. Our strategy is to generalize the forward propagation of a Graph Convolutional Network (GCN), which is a popular graph NN variant, as a specific dynamical system. In the case of a GCN, we show that when its weights satisfy the conditions determined by the spectra of the (augmented) normalized Laplacian, its output exponentially approaches the set of signals that carry information of the connected components and node degrees only for distinguishing nodes. Our theory enables us to relate the expressive power of GCNs with the topological information of the underlying graphs inherent in the graph spectra. To demonstrate this, we characterize the asymptotic behavior of GCNs on the Erdős – Rényi graph. We show that when the Erdős – Rényi graph is sufficiently dense and large, a broad range of GCNs on it suffers from the "information loss" in the limit of infinite layers with high probability. Based on the theory, we provide a principled guideline for weight normalization of graph NNs. We experimentally confirm that the proposed weight scaling enhances the predictive performance of GCNs in real data[1].

## 1 Introduction

Motivated by the success of Deep Learning (DL), several attempts have been made to apply DL models to non-Euclidean data, particularly, graph-structured data such as chemical compounds, social networks, and polygons. Recently, *Graph Neural Networks* (graph NNs) (Duvenaud et al., 2015; Li et al., 2016; Gilmer et al., 2017; Hamilton et al., 2017; Kipf & Welling, 2017; Nguyen et al., 2017; Schlichtkrull et al., 2018; Battaglia et al., 2018; Xu et al., 2019; Wu et al., 2019a) have emerged as a promising approach. However, despite their practical popularity, theoretical research of graph NNs has not been explored extensively.

The characterization of DL model *expressive power*, i.e., to identify what function classes DL models can (approximately) represent, is a fundamental question in theoretical research of DL. Many studies have been conducted for Fully Connected Neural Networks (FNNs) (Cybenko, 1989; Hornik, 1991; Hornik et al., 1989; Barron, 1993; Mhaskar, 1993; Sonoda & Murata, 2017; Yarotsky, 2017) and Convolutional Neural Networks (CNNs) (Petersen & Voigtlaender, 2018; Zhou, 2018; Oono & Suzuki, 2019). For such models, we have theoretical and empirical justification that deep and non-linear architectures can enhance representation power (Telgarsky, 2016; Chen et al., 2018b; Zhou & Feng, 2018). However, for graph NNs, several papers have reported that node representations go indistinguishable (known as *over-smoothing*) and prediction performances severely degrade when we stack many layers (Kipf & Welling, 2017; Wu et al., 2019b; Li et al., 2018). Besides, Wu et al. (2019a) reported that graph NNs achieved comparable performance even if they removed intermediate non-linear functions. These studies posed a question about the current architecture and made us aware of the need for the theoretical analysis of the graph NN expressive power.

---

[1]Code is available at https://github.com/delta2323/gnn-asymptotics.

In this paper, we investigate the expressive power of graph NNs by analyzing their asymptotic behaviors as the layer size goes to infinity. Our theory gives new theoretical conditions under which neither layer stacking nor non-linearity contributes to improving expressive power. We consider a specific dynamics that includes a transition defining a Markov process and the forward propagation of a Graph Convolutional Network (GCN) (Kipf & Welling, 2017), which is one of the most popular graph NN variants, as special cases. We prove that under certain conditions, the dynamics exponentially approaches a subspace that is invariant under the dynamics. In the case of GCN, the invariant space is a set of signals that correspond to the lowest frequency of graph spectra and that have "no information" other than connected components and node degrees for a node classification task whose goal is to predict the nodes' properties in a graph. The rate of the distance between the output and the invariant space is $O((s\lambda)^L)$ where $s$ is the maximum singular values of weights, $\lambda$ is typically a quantity determined by the spectra of the (augmented) normalized Laplacian, and $L$ is the layer size. See Sections 3.3 (general case) and 4 (GCN case) for precise statements.

We can interpret our theorem as the generalization of the well-known property that if a finite and discrete Markov process is irreducible and aperiodic, it exponentially converges to a unique equilibrium and the eigenvalues of its transition matrix determine the convergence rate (see, e.g., Chung & Graham (1997)). Different from the Markov process case, which is linear, the existence of intermediate non-linear functions complicates the analysis. We overcame this problem by leveraging the combination of the ReLU activation function (Krizhevsky et al., 2012) and the positivity of eigenvectors of the Laplacian associated with the smallest positive eigenvalues.

Our theory enables us to investigate asymptotic behaviors of graph NNs via the spectral distribution of the underlying graphs. To demonstrate this, we take GCNs defined on the Erdős – Rényi graph $G_{N,p}$, which has $N$ nodes and each edge appears independently with probability $p$, for an example. We prove that if $\frac{\log N}{pN} = o(1)$ as a function of $N$, any GCN whose weights have maximum singular values at most $C\sqrt{\frac{Np}{\log(N/\varepsilon)}}$ approaches the "information-less" invariant space with probability at least $1 - \varepsilon$, where $C$ is a universal constant. Intuitively, if the graph on which we define graph NNs is sufficiently dense, graph-convolution operations mix signals on nodes fast and hence the feature maps lose information for distinguishing nodes quickly.

Our contributions are as follows:

- We relate asymptotic behaviors of graph NNs with the topological information of underlying graphs via the spectral distribution of the (augmented) normalized Laplacian.
- We prove that if the weights of a GCN satisfy conditions determined by the graph spectra, the output of the GCN carries no information other than the node degrees and connected components for discriminating nodes when the layer size goes to infinity (Theorems 1, 2).
- We apply our theory to Erdős – Rényi graphs as an example and show that when the graph is sufficiently dense and large, many GCNs suffer from the information loss (Theorem 3).
- We propose a principled guideline for weight normalization of graph NNs and empirically confirm it using real data.

## 2 RELATED WORK

**MPNN-type Graph NNs.** Since many graph NN variants have been proposed, there are several unified formulations of graph NNs (Gilmer et al., 2017; Battaglia et al., 2018). Our approach is the closest to the formulation of Message Passing Neural Network (MPNN) (Gilmer et al., 2017), which unified graph NNs in terms of the update and readout operations. Many graph NNs fall into this formulation such as Duvenaud et al. (2015), Li et al. (2016), and Veličković et al. (2018). Among others, GCN (Kipf & Welling, 2017) is an important application of our theory because it is one of the most widely used graph NNs. In addition, GCNs are interesting from a theoretical research perspective because, in addition to an MPNN-type graph NN, we can interpret GCNs as a simplification of spectral-type graph NNs (Henaff et al., 2015; Defferrard et al., 2016), that make use of the graph Laplacian.

Our approach, which considers the asymptotic behaviors graph NNs as the layer size goes to infinity, is similar to Scarselli et al. (2009), one of the earliest works about graph NNs. They obtained node

representations by iterating message passing between nodes until convergence. Their formulation is general in that we can use any local aggregation operation as long as it is a *contraction map*. Our theory differs from theirs in that we proved that the output of a graph NN approaches a certain space even if the local aggregation function is not necessarily a contraction map.

**Expressive Power of Graph NNs.** Several studies have focused on theoretical analysis and the improvement of graph NN expressive power. For example, Xu et al. (2019) proved that graph NNs are no more powerful than the Weisfeiler – Lehman (WL) isomorphism test (Weisfeiler & A.A., 1968) and proposed a Graph Isomorphism Network (GIN), that is approximately as powerful as the WL test. Although they experimentally showed that GIN has improved accuracy in supervised learning tasks, their analysis was restricted to the graph isomorphism problem. Xu et al. (2018) analyzed the non-asymptotic properties of GCNs through the lens of random walk theory. They proved the limitations of GCNs in expander-like graphs and proposed a Jumping Knowledge Network (JK-Net) to address the issue. To handle the non-linearity, they linearized networks by a randomization assumption (Choromanska et al., 2015). We take a different strategy and make use of the interpretation of ReLU as a projection onto a cone. Recently, NT & Maehara (2019) showed that a GCN approximately works as a low-pass filter plus an MLP in a certain setting. Although they analyzed finite-depth GCNs, our theory has similar spirits with theirs because our "information-less" space corresponds to the lowest frequency of a graph Laplacian. Another point is that they imposed assumptions that input signals consist of low-frequent true signals and high-frequent noise, whereas we need not such an assumption.

**Role of Deep and Non-linear Structures.** For ordinal DL models such as FNNs and CNNs, we have both theoretical and empirical justification of deep and non-linear architectures for enhancing of the expressive power (e.g., Telgarsky (2016); Petersen & Voigtlaender (2018); Oono & Suzuki (2019)). In contrast, several studies have witnessed severe performance degradation when stacking many layers on graph NNs (Kipf & Welling, 2017; Wu et al., 2019b). Li et al. (2018) reported that feature vectors on nodes in a graph go indistinguishable as we increase layers in several tasks. They named this phenomenon *over-smoothing*. Regarding non-linearity, Wu et al. (2019a) empirically showed that graph NNs achieve comparable performance even if we omit intermediate non-linearity. These observations gave us questions about the current models of deep graph NNs in terms of their expressive power. Several studies gave theoretical explanations of the over-smoothing phenomena for *linear* GNNs (Li et al., 2018; Zhang, 2019; Zhao & Akoglu, 2020). We can think of our theory as an extension of their results to non-linear GNNs.

## 3 Problem Setting and Main Result

### 3.1 Notation

Let $\mathbb{N}_+$ be the set of positive integers. For $N \in \mathbb{N}_+$, we denote $[N] := \{1, \ldots, N\}$. For a vector $v \in \mathbb{R}^N$, we write $v \geq 0$ if and only if $v_n \geq 0$ for all $n \in [N]$. Similarly, for a matrix $X \in \mathbb{R}^{N \times C}$, we write $X \geq 0$ if and only if $X_{nc} \geq 0$ for all $n \in [N]$ and $c \in [C]$. We say such a vector and matrix is *non-negative*. $\langle \cdot, \cdot \rangle$ denotes the inner product of vectors or matrices, depending on the context: $\langle u, v \rangle := u^\top v$ for $u, v \in \mathbb{R}^N$ and $\langle X, Y \rangle := \mathrm{tr}(X^T Y)$ for $X, Y \in \mathbb{R}^{N \times C}$. $\mathbf{1}_P$ equals to 1 if the proposition $P$ is true else 0. For vectors $v \in \mathbb{R}^N$ and $w \in \mathbb{R}^C$, $v \otimes w \in \mathbb{R}^{N \times C}$ denotes the Kronecker product of $v$ and $w$ defined by $(v \otimes w)_{nc} := v_n w_c$. For $X \in \mathbb{R}^{N \times C}$, $\|X\|_{\mathrm{F}} := \langle X, X \rangle^{\frac{1}{2}}$ denotes the Frobenius norm of $X$. For a vector $v \in \mathbb{R}^N$, $\mathrm{diag}(v) := (v_n \delta_{nm})_{n,m \in [N]} \in \mathbb{R}^{N \times N}$ denotes the diagonalization of $v$. $I_N \in \mathbb{R}^{N \times N}$ denotes the identity matrix of size $N$. For a linear operator $P : \mathbb{R}^N \to \mathbb{R}^M$ and a subset $V \subset \mathbb{R}^N$, we denote the restriction of $P$ to $V$ by $P|_V$.

### 3.2 Dynamical System

Although we are mainly interested in GCNs, we develop our theory more generally using dynamical systems. We will specialize to the GCNs in Section 4.

For $N, C, H_l \in \mathbb{N}_+$ ($l \in \mathbb{N}_+$), let $P \in \mathbb{R}^{N \times N}$ be a symmetric matrix and $W_{lh} \in \mathbb{R}^{C \times C}$ for $l \in \mathbb{N}_+$ and $h \in [H_l]$. We define $f_l : \mathbb{R}^{N \times C} \to \mathbb{R}^{N \times C}$ by $f_l(X) := \mathrm{MLP}_l(PX)$. Here, $\mathrm{MLP}_l : \mathbb{R}^{N \times C} \to \mathbb{R}^{N \times C}$ is the $l$-th multi-layer perceptron common to all nodes (Xu et al., 2019) and is defined by $\mathrm{MLP}_l(X) := \sigma(\cdots \sigma(\sigma(X)W_{l1})W_{l2} \cdots W_{lH_l})$, where $\sigma : \mathbb{R}^{N \times C} \to \mathbb{R}^{N \times C}$

is an element-wise ReLU function (Krizhevsky et al., 2012) defined by $\sigma(X)_{nc} := \max(X_{nc}, 0)$ for $n \in [N], c \in [C]$. We consider the dynamics $X^{(l+1)} := f_l(X^{(l)})$ with some initial value $X^{(0)} \in \mathbb{R}^{N \times C}$. We are interested in the asymptotic behavior of $X^{(l)}$ as $l \to \infty$.

For $M \leq N$, let $U$ be a $M$-dimensional subspace of $\mathbb{R}^N$. We assume that $U$ and $P$ satisfy the following properties that generalize the situation where $U$ is the eigenspace associated with the smallest eigenvalue of a (normalized) graph Laplacian $\Delta$ (that is, zero) and $P$ is a polynomial of $\Delta$.

**Assumption 1.** *$U$ has an orthonormal basis $(e_m)_{m \in [M]}$ that consists of non-negative vectors.*

**Assumption 2.** *$U$ is invariant under $P$, i.e., if $u \in U$, then $Pu \in U$.*

We endow $\mathbb{R}^N$ with the ordinal inner product and denote the orthogonal complement of $U$ by $U^\perp := \{u \in \mathbb{R}^N \mid \langle u, v \rangle = 0, \forall v \in U\}$. By the symmetry of $P$, we can show that $U^\perp$ is invariant under $P$, too (Appendix E.1, Proposition 2). Therefore, we can regard $P$ as a linear mapping $P|_{U^\perp} : U^\perp \to U^\perp$. We denote the operator norm of $P|_{U^\perp}$ by $\lambda$. When $U$ is the eigenspace associated with the smallest eigenvalue of $\Delta$ and $P$ is $g(\Delta)$ where $g$ is a polynomial, then, $\lambda$ corresponds to $\lambda = \sup_\mu |g(\mu)|$ where $\sup$ ranges over all eigenvalues except the smallest one.

### 3.3 MAIN RESULT

We define the subspace $\mathcal{M}$ of $\mathbb{R}^{N \times C}$ by $\mathcal{M} := U \otimes \mathbb{R}^C = \{\sum_{m=1}^M e_m \otimes w_m \mid w_m \in \mathbb{R}^C\}$ where $(e_m)_{m \in [M]}$ is the orthonormal basis of $U$ appeared in Assumption 1. For $X \in \mathbb{R}^{N \times C}$, we denote the distance between $X$ and $\mathcal{M}$ by $d_{\mathcal{M}}(X) := \inf\{\|X - Y\|_F \mid Y \in \mathcal{M}\}$. We denote the maximum singular value of $W_{lh}$ by $s_{lh}$ and set $s_l := \prod_{h=1}^{H_l} s_{lh}$. With these preparations, we introduce the main theorem of the paper.

**Theorem 1.** *Under Assumptions 1 and 2, we have $d_{\mathcal{M}}(f_l(X)) \leq s_l \lambda d_{\mathcal{M}}(X)$ for any $X \in \mathbb{R}^{N \times C}$.*

The proof key is that the non-linear operation $\sigma$ decreases the distance $d_{\mathcal{M}}$, that is, $d_{\mathcal{M}}(\sigma(X)) \leq d_{\mathcal{M}}(X)$. We use the non-negativity of $e_m$ to prove this claim. See Appendix A for the complete proof. We also discuss the strictness of Theorem 1 in Appendix E.3.

By setting $d_{\mathcal{M}}(X) = 0$, this theorem implies that $\mathcal{M}$ is invariant under $f_l$. In addition, if the maximum value of singular values are small, $X^{(l)}$ asymptotically approaches $\mathcal{M}$ in the sense of Johnson (1973) for any initial value $X^{(0)}$. That is, the followings hold under Assumptions 1 and 2.

**Corollary 1.** *$\mathcal{M}$ is invariant under $f_l$ for any $l \in \mathbb{N}_+$, that is, if $X \in \mathcal{M}$, then we have $f_l(X) \in \mathcal{M}$.*

**Corollary 2.** *Let $s := \sup_{l \in \mathbb{N}_+} s_l$. We have $d_{\mathcal{M}}(X^{(l)}) = O((s\lambda)^l)$. In particular, if $s\lambda < 1$, then $X_l$ exponentially approaches $\mathcal{M}$ as $l \to \infty$ for any initial value $X^{(0)}$.*

Suppose the operator norm of $P|_U : U \to U$ is no larger than $\lambda$, then, under the assumption of $s\lambda < 1$, $X^{(l)}$ converges to 0, the trivial fixed point (see Appendix E.2, Proposition 3). Therefore, we are mainly interested in the case where the operator norm of $P|_U$ is strictly larger than $\lambda$ (see Proposition 1). Finally, we restate Theorem 1 specialized to the situation where $U$ is the direct sum of eigenspaces associated with the largest $M$ eigenvalues of $P$. Note that the eigenvalues of $P$ is real since $P$ is symmetric.

**Corollary 3.** *Let $\lambda_1 \leq \cdots \leq \lambda_N$ be the eigenvalue of $P$, sorted in ascending order. Suppose the multiplicity of the largest eigenvalue $\lambda_N$ is $M(\leq N)$, i.e., $\lambda_{N-M} < \lambda_{N-M+1} = \cdots = \lambda_N$. We define $\lambda := \max_{n \in [N-M]} |\lambda_n|$. We denote $U$ by the eigenspace associated with $\lambda_N$ and assume that $U$ satisfies Assumption 1. Then, we have $d_{\mathcal{M}}(X^{(l+1)}) \leq s_l \lambda d_{\mathcal{M}}(X^{(l)})$.*

**Remark 1.** *It is known that any Markov process on finite states converges to a unique distribution (equilibrium) if it is irreducible and aperiodic (see e.g., Norris (1998)). Theorem 1 includes this proposition as a special case with $M = 1$, $C = 1$, and $W_l = 1$ for all $l \in \mathbb{N}_+$. This is essentially the direct consequence of Perron – Frobenius' theorem (see e.g., Meyer (2000)). See Appendix F.*

## 4 APPLICATION TO GCN

We formulate a GCN (Kipf & Welling, 2017) without readout operations (Gilmer et al., 2017) using the dynamical system in the previous section and derive a sufficient condition in terms of the spectra of underlying graphs in which layer stacking nor non-linearity are not helpful for node classification.

Let $G = (V, E)$ be an undirected graph where $V$ is a set of nodes and $E$ is a set of edges. We denote the number of nodes in $G$ by $N = |V|$ and identify $V$ with $[N]$ by fixing an order of $V$. We associate a $C$ dimensional signal to each node. $X$ in the previous section corresponds to concatenation of the signals. GCNs iteratively update signals on $V$ using the connection information and weights.

Let $A := (\mathbf{1}_{\{(i,j) \in E\}})_{i,j \in [N]} \in \mathbb{R}^{N \times N}$ be the adjacency matrix and $D := \mathrm{diag}(\deg(i)_{i \in [N]}) \in \mathbb{R}^{N \times N}$ be the degree matrix of $G$ where $\deg(i) := |\{j \in V \mid (i,j) \in E\}|$ is the degree of node $i$. Let $\tilde{A} := A + I_N$, $\tilde{D} := D + I_N$ be the adjacent and degree matrix of graph $G$ augmented with self-loops. We define the *augmented* normalized Laplacian (Wu et al., 2019a) of $G$ by $\tilde{\Delta} := I_N - \tilde{D}^{-\frac{1}{2}} \tilde{A} \tilde{D}^{-\frac{1}{2}}$ and set $P := I_N - \tilde{\Delta}$. Let $L, C \in \mathbb{N}_+$ be the layer and channel sizes, respectively. For weights $W_l \in \mathbb{R}^{C \times C}$ ($l \in [L]$), we define a GCN[2] associated with $G$ by $f = f_L \circ \cdots \circ f_1$ where $f_l : \mathbb{R}^{N \times C} \to \mathbb{R}^{N \times C}$ is defined by $f_l(X) := \sigma(PXW_l)$. We are interested in the asymptotic behavior of the output $X^{(L)}$ of the GCN as $L \to \infty$.

Suppose $G$ has $M$ connected components and let $V = V_1 \sqcup \cdots \sqcup V_M$ be the decomposition of the node set $V$ into connected components. We denote an indicator vector of the $m$-th connected component by $u_m := (\mathbf{1}_{\{n \in V_m\}})_{n \in [N]} \in \mathbb{R}^N$. The following proposition shows that GCN satisfies the assumption of Corollay 3 (see Appendix B for proof).

**Proposition 1.** *Let $\lambda_1 \leq \cdots \leq \lambda_N$ be the eigenvalue of $P$ sorted in ascending order. Then, we have $-1 < \lambda_1$, $\lambda_{N-M} < 1$, and $\lambda_{N-M+1} = \cdots = \lambda_N = 1$. In particular, we have $\lambda := \max_{n=1,\ldots,N-M} |\lambda_n| < 1$. Further, $e_m := \tilde{D}^{\frac{1}{2}} u_m$ for $m \in [M]$ are the basis of the eigenspace associated with the eigenvalue $1$.*

**Theorem 2.** *For any initial value $X^{(0)}$, the output of $l$-th layer $X^{(l)}$ satisfies $d_{\mathcal{M}}(X^{(l)}) \leq (s\lambda)^l d_{\mathcal{M}}(X^{(0)})$. In particular, $d_{\mathcal{M}}(X^{(l)})$ exponentially converges to $0$ when $s\lambda < 1$.*

In the context of node classification tasks, we can interpret this corollary as the "information loss" of GCNs in the limit of infinite layers. For any $X \in \mathcal{M}$, if two nodes $i, j \in V$ are in a same connected component and their degrees are identical, then, the column vectors of $X$ that correspond to nodes $i$ and $j$ are identical. It means that we cannot distinguish these nodes using $X$. In this sense, $\mathcal{M}$ only has information about connected components and node degrees and we can interpret this theorem as the exponential information loss of GCNs in terms of the layer size. Similarly to the discussion in the previous section, $X^{(l)}$ converges to the trivial fixed point $0$ when $s < 1$ (remember $\lambda_N = 1$). An interesting point is that even if $s \geq 1$, $X^{(l)}$ can suffer from this information loss when $s < \lambda^{-1}$.

We note that the rate $s\lambda$ in Theorem 2 depends on the spectra of the augmented normalized Laplacian, which is determined by the topology of the underlying graph $G$. Hence, our result explicitly relates the topological information of graphs and asymptotic behaviors of graph NNs.

**Remark 2.** *The old preprint (version 2) of Luan et al. (2019) formulated a theorem that explains the over-smoothing of non-linear GNNs. Specifically, it claimed that if a graph does not have a bipartite component and the input distribution is continuous, the rank of the output of a GCN converges to the number of connected components of the underlying graph as the layer size goes to infinity almost surely. However, it is not true in general as we give a counterexample in Appendix C.*

## 5 ASYMPTOTIC BEHAVIOR OF GCN ON ERDŐS – RÉNYI GRAPH

Theorem 2 gives us a way to characterize the asymptotic behaviors of GCNs via the spectral distributions of the underlying graphs. To demonstrate this, we consider an Erdős – Rényi graph $G_{N,p}$ (Erdös & Rényi, 1959; Gilbert, 1959), which is a random graph that has $N$ nodes and whose edges between two distinct nodes appear independently with probability $p \in [0, 1]$, as an example. First, consider a (non-random) graph $G$ with $M$ connected components. Let $0 = \tilde{\mu}_1 = \cdots = \tilde{\mu}_M < \tilde{\mu}_{M+1} \leq \cdots \leq \tilde{\mu}_N < 2$ be eigenvalues of the augmented normalized Laplacian of $G$ (see, Proposition 1) and set $\lambda := \min_{m=M+1,\ldots,N} |1 - \tilde{\mu}_m| (< 1)$. By Theorem 2, the output of GCN "loses information" as the layer size goes to infinity when the largest singular values of weights are strictly smaller than $\lambda^{-1}$. Therefore, the closer the positive eigenvalues $\mu_m$ are to $1$, the broader range of GCNs satisfies the assumption of Theorem 2.

---

[2]Following the original paper (Kipf & Welling, 2017), we use one-layer MLPs (i.e., $H_l = 1$ for all $l \in \mathbb{N}_+$.). However, our result holds for the multi-layer case

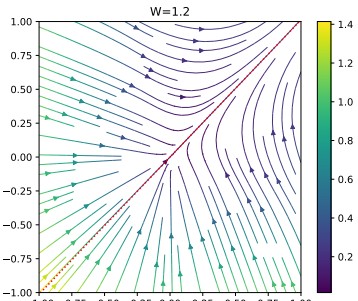 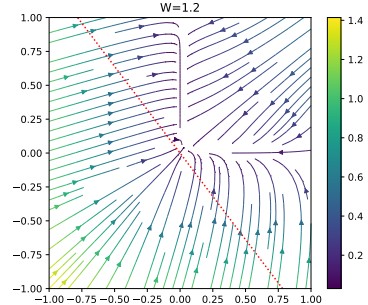

Figure 1: Visualization of vector field $V(X) := f(X) - X$ induced by the one-step transition. Color maps indicate the absolute value $|V(X)|$ at the point $X$. Dotted lines are the subspace $\mathcal{M}$. Left: **Case 1**. Right: **Case 2**. Best view in color.

For an Erdős – Rényi graph $G_{N,p}$, Chung & Radcliffe (2011) showed that when $\frac{\log N}{Np} = o(1)$, the eigenvalues of the (usual) normalized Laplacian except for the smallest one converge to 1 with high probability (see Theorem 2 therein)[3]. We can interpret this theorem as the convergence of Erdős-Rényi graphs to the complete graph in terms of graph spectra. We can prove that the augmented normalized Laplacian behaves similarly (Lemma 6). By combining this fact with the discussion in the previous paragraph, we obtain the asymptotic behavior of GCNs on the Erdős – Rényi graph. See Appendix D for the complete proof.

**Theorem 3.** *Consider a GCN on the Erdős-Rényi graph $G_{N,p}$ such that $\frac{\log N}{Np} = o(1)$ as a function of $N$. For any $\varepsilon > 0$, if the supremum $s$ of the maximum singular values of weights in the GCN satisfies $s < s_0 := \frac{1}{7}\sqrt{\frac{Np - p + 1}{\log(4N/\varepsilon)}}$, then, for sufficiently large $N$, the GCN satisfies the condition of Theorem 2 with probability at least $1 - \varepsilon$.*

Theorem 3 requires that an underlying graph is not extremely sparse. For example, suppose the node size is $N = 20,000$, which is the approximately the maximum node size of datasets we use in experiments, and the edge probability is $p = \log N / N$. Then, each node has the order of $Np \approx 4.3$ adjacent nodes.

Under the condition of Theorem 3, the upper bound $s_0 \to \infty$ as $N \to \infty$. It means that if the graph is sufficiently large and not extremely sparse, most GCNs suffer from the information loss. For the dependence on the edge probability $p$, $s_0$ is an increasing function of $p$, which means the denser a graph is, the more quickly graph convolution operations mix signals on nodes and move them close to each other.

Theorem 3 implies that graph NNs perform poorly on dense NNs. More aggressively, we can hypothesize that the sparsity of practically available graphs is one of the reasons for the success of graph NNs in node classification tasks. To confirm this hypothesis, we artificially add edges to citation networks to make them dense in the experiments and observe the failure of graph NNs as expected (see Section 6.3).

## 6 EXPERIMENT

### 6.1 SYNTHESIS DATA: ONE-STEP TRANSITION

We numerically investigate how the transition $f(X) := \sigma(PXW)$ changes inputs using the vector field $V(X) := f(X) - X$[4]. For this purpose, we set $N = 2$, $M = 1$, and $C = 1$. Let $\lambda_1 \leq \lambda_2$ be the eigenvalues of $P$. We choose $W$ as $|\lambda_2|^{-1} \leq W < |\lambda_1|^{-1}$ so that Theorem 1 is applicable but is not reduced to the trivial situation (see, Appendix E.2). We choose the eigenvector $e \in \mathbb{R}^2$

---

[3]Chung et al. (2004) and Coja-Oghlan (2007) proved similar theorems.

[4]Since we consider the one-step transition only, we omit the subscript $l$ from $f_l$, $X_l$, and $W_l$.

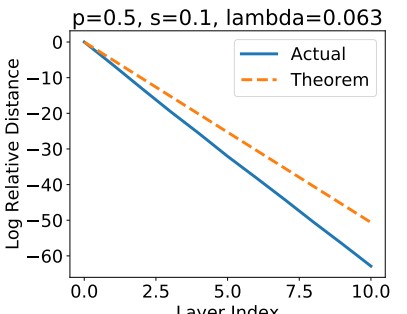 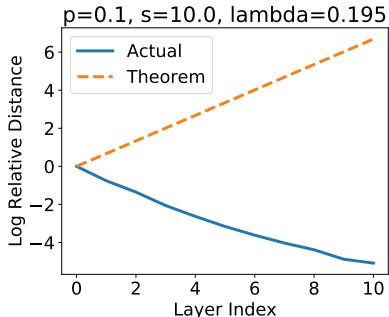

Figure 2: The actual distances to the invariant space $\mathcal{M}$ and their upper bounds. Solid lines are the log relative distance defined by $y(l) = \log(d_{\mathcal{M}}(X^{(l)})/d_{\mathcal{M}}(X^{(0)}))$ and dotted lines are upper bound $y(l) = l \log(s\lambda)$, where $X^{(0)}$ is the input signal and $X^{(l)}$ is the output of the $l$-th layer.

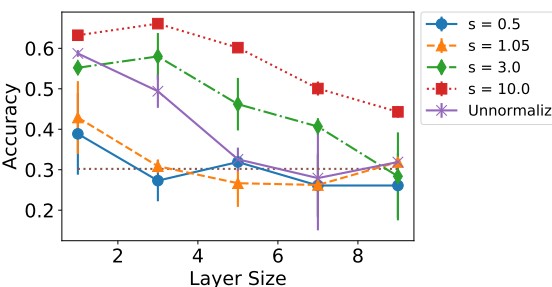 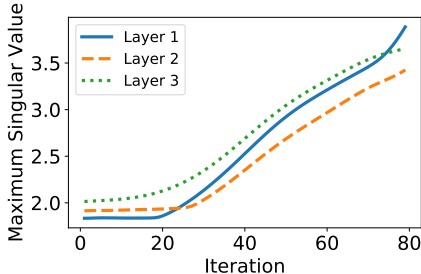

Figure 3: Node prediction results on Noisy Cora. Left: Effect of the maximum singular values on weights on model performance. The horizontal dotted line indicates the chance rate (30.2%). The error bar is the standard deviation of 3 trials. Right: Transition of maximum singular values during training. See Appendix I.3 for results using other datasets. Best view in color.

associated with $\lambda_2$ in two ways as described below. See Appendix H.1 for the concrete values of $P$, $e$, and $W$. Figure 1 shows the visualization of $V$. First, we choose the non-negative eigenvector $e$ so that it satisfies Assumption 1 (**Case 1**). We see that the transition function $f$ uniformly decreases the distance from $\mathcal{M}$. This is consistent with the consequence of Theorem 1. Next, we choose the eigenvector $e = \begin{bmatrix} e_1 & e_2 \end{bmatrix}^{\top}$ such that the signs of $e_1$ and $e_2$ differ (**Case 2**), which violates Assumption 1. We see that $\mathcal{M}$ is not invariant under $f$ and $f$ does not uniformly decrease the distance from $\mathcal{M}$. Therefore, we cannot remove the non-negativity assumption from Theorem 1.

## 6.2 Synthesis Data: Distance to Invariant Space

We evaluate the distance to the invariant space $\mathcal{M}$ using synthesis data. We randomly generate an Erdős–Rényi graph, a GCN on it, and an input signal $X^{(0)}$. We compute the distance between the $l$-th intermediate output $X^{(l)}$ and the invariant space $\mathcal{M}$ for various edge probability $p$ and maximum singular value $s$. Figure 2 plots the logarithm of the relative distance $y(l) = \log \frac{d_{\mathcal{M}}(X^{(l)})}{d_{\mathcal{M}}(X^{(0)})}$ with respect to the layer index $l$. From Theorem 1, we know that it is upper bounded by $y(l) = l \log(s\lambda)$. We see that this bound well approximates the actual value when $s\lambda$ is small. On the other hand, it is loose for large $s\lambda$. We leave tighter bounds for $d_{\mathcal{M}}$ in such a case for future research.

## 6.3 Real Data: Effect of Maximum Singular Values on Performance

Theorem 2 implies that if $s$ is smaller than the threshold $\lambda^{-1}$, we cannot expect deep GCN to achieve good prediction accuracy. Conversely, if we can successfully train the model, $s$ should avoid the region $s \leq \lambda^{-1}$. We empirically confirm these hypotheses using real datasets.

We use Cora, CiteSeer, and PubMed (Sen et al., 2008), which are standard citation network datasets. The task is to classify the genre of papers using word occurrences and citation relationships. We regard each paper as a node and citation relationship as an edge. Due to space constraints, we focus on Cora in the main article. See Appendix H.3 and I.3 for the other datasets. The discussion in Section 5 implies that Theorem 2 can support a wide range of GCNs when the underlying graph is relatively dense. However, the citation networks are too sparse to examine the aforementioned hypotheses — Theorem 2 gives a non-trivial result only when $1 \leq s < \lambda^{-1} \approx 1 + 3.62 \times 10^{-3}$. To circumvent this, we make *noisy versions* of citation networks by randomly adding edges to graphs. Through this manipulation, we can increase the value of $\lambda^{-1}$ to 1.11.

Figure 3 (left) shows the accuracy for the test dataset in terms of the maximum singular values and the number of graph convolution layers. We can observe that when GCNs whose maximum singular value $s$ is out of the region $s < \lambda^{-1}$ outperform those inside the region in almost all configurations. Furthermore, the accuracy of GCNs with $s = 10$ are better than those without normalization (*unnormalized*). Figure 3 (right) shows the transition of the maximum singular values of the weights during training when we use a three-layered GCN. We can observe that the maximum singular value $s$ does not shrink to the region $s \leq \lambda^{-1}$. In addition, when the layer size is small and predictive accuracy is high, GCNs gradually increase $s$ from the initial value and avoid the region. In conclusion, the experiment results are consistent with the theorems.

## 6.4 REAL DATA: EFFECT OF SIGNAL COMPONENT PERPENDICULAR TO INVARIANT SPACE

We can decompose the output $X$ of a model as $X = X_0 + X_1$ ($X_0 \in \mathcal{M}$, $X_1 \in \mathcal{M}^\perp$). According to the theory, $X_0$ has limited information for node classification. We hypothesize that the model emphasizes the perpendicular component $X_1$ to perform good predictions. To quantitatively evaluate it, we define the relative magnitude of the perpendicular component of the output $X$ by $t(X) := X_1/X_0$. Figure 4 compares this quantity and the prediction accuracy on the noisy version of Cora (see Appendix I.4 for other datasets). We observe that these two quantities are correlated ($R = 0.545$). If we remove GCNs have only one layer (corresponding to right points in the figure), the correlation coefficient is $0.827$. This result does not contradict to the hypothesis above [5].

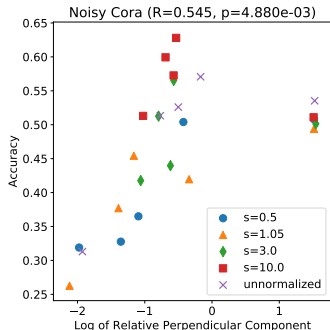

Figure 4: $\log t(X)$ and prediction accuracy on Noisy Cora.

## 7 DISCUSSION

**Applicability to Graph NNs on Sparse Graphs.** We have theoretically and empirically shown that when the underlying graph is sufficiently dense and large, the threshold $\lambda^{-1}$ is large (Theorem 2 and Section 6.3), which means many graph CNNs are eligible. However, real-world graphs are not often dense, which means that Theorem 2 is applicable to very limited GCNs. In addition, Coja-Oghlan (2007) theoretically proved that if the expected average degree of $G_{N,p}$ is bounded, the smallest positive eigenvalue of the normalized Laplacian of $G_{N,p}$ is $o(1)$ with high probability. The asymptotic behaviors of graph NNs on sparse graphs are left for future research.

**Remedy for Over-smoothing.** Based on our theory, we can propose several techniques for mitigating the over-smoothing phenomena. One idea is to (randomly) sample edges in an underlying graph. The sparsity of practically available graphs could be a factor in the success of graph NNs. Assuming this hypothesis is correct, there is a possibility that we can relive the effect of over-smoothing by sparsification. Since we can never restore the information in pruned edges if we remove them permanently, random edge sampling could work better as FastGCN (Chen et al., 2018a) and Graph-SAGE (Hamilton et al., 2017) do. Another idea is to scale node representations (i.e., intermediate or final output of graph NNs) appropriately so that they keep away from the invariant space $\mathcal{M}$. Our proposed weight scaling mechanism takes this strategy. Recently, Zhao & Akoglu (2020) has pro-

---

[5]We cannot conclude that large perpendicular components are essential for good performance, since the maximum singular value $s$ is correlated to the accuracy, too.

posed PairNorm to alleviate the over-smoothing phenomena. Although the scaling target is different – they rescaled signals whereas we normalized weights – theirs and ours have similar spirits.

**Graph NNs with Large Weights.** Our theory suggests that the maximum singular values of weights in a GCN should not be smaller than a threshold $\lambda^{-1}$ because it suffers from information loss for node classification. On the other hand, if the scale of weights are very large, the model complexity of the function class represented by graph NNs increases, which may cause large generalization errors. Therefore, from a statistical learning theory perspective, we conjecture that the graph NNs with too-large weights perform poorly, too. A trade-off should exist between the expressive power and model complexity and there should be a "sweet spot" on the weight scale that balances the two.

**Relation to Double Descent Phenomena.** Belkin et al. (2019) pointed out that modern deep models often have *double descent* risk curves: when a model is under-parameterized, a classical bias-variance trade-off occurs. However, once the model has a large capacity and perfectly fits the training data, the test error decreases as we increase the number of parameters. To the best of our knowledge, no literature reported the double descent phenomena for graph NNs (it is consistent with the picture of the classical U-shaped risk curve in the previous paragraph). It is known that double descent phenomena do not occur in some situations, especially depending on regularization types. For example, while Belkin et al. (2019) employed the interpolating hypothesis with the minimum norm, Mei & Montanari (2019) found that the double descent was alleviated or disappeared when they used Ridge-type regularization techniques. Therefore, one can hypothesize the over-smoothing is a cause or consequence of regularization that is more like a Ridge-type rather than minimum-norm inductive bias.

**Limitations in Graph NN Architectures.** Our analysis is limited to graph NNs with the ReLU activation function because we implicitly use the property that ReLU is a projection onto the cone $\{X \geq 0\}$ (Appendix A, Lemma 3). This fact enables the ReLU function to get along with the non-negativity of eigenvectors associated with the largest eigenvalues. Therefore, it is far from trivial to extend our results to other activation functions such as the sigmoid function or Leaky ReLU (Maas et al., 2013). Another point is that our formulation considers the update operation (Gilmer et al., 2017) of graph NNs only and does not take readout operations into account. In particular, we cannot directly apply our theory to graph classification tasks in which each sample is a graph.

**Over-smoothing of Residual GNNs.** Considering the correspondence of graph NNs and Markov processes (see Appendix F), one can imagine that residual links do not contribute to alleviating the over-smoothing phenomena because adding residual connections to a graph NN corresponds to converting a Markov process to its lazy version. When a Markov process converges to a stable distribution, the corresponding lazy process also converges eventually under certain conditions. It implies that residual links might not be helpful. However, Li et al. (2019) reported that graph NNs with as many as 56 layers performed well if they added residual connections. Considering that, the situation could be more complicated than our intuitions. The analysis of the role of residual connections in graph NNs is a promising direction for future research.

# 8 Conclusion

In this paper, to understand the empirically observed phenomena that deep non-linear graph NNs do not perform well, we analyzed their asymptotic behaviors by interpreting them as a dynamical system that includes GCN and Markov process as special cases. We gave theoretical conditions under which GCNs suffer from the information loss in the limit of infinite layers. Our theory directly related the expressive power of graph NNs and topological information of the underlying graphs via spectra of the Laplacian. It enabled us to leverage spectral and random graph theory to analyze the expressive power of graph NNs. To demonstrate this, we considered GCN on the Erdős – Rényi graph as an example and showed that when the underlying graph is sufficiently dense and large, a wide range of GCNs on the graph suffer from information loss. Based on the theory, we gave a principled guideline for how to determine the scale of weights of graph NNs and empirically showed that the weight normalization implied by our theory performed well in real datasets. One promising direction of research is to analyze the optimization and statistical properties such as the generalization power (Verma & Zhang, 2019) of graph NNs via spectral and random graph theories.

ACKNOWLEDGMENTS

We thank Katsuhiko Ishiguro for providing a part of code for the experiments, Kohei Hayashi and Haru Negami Oono for giving us feedback and comments on the draft, Keyulu Xu and anonymous reviewers for fruitful discussions via OpenReview, and Ryuta Osawa for pointing out errors and suggesting improvements of the paper. TS was partially supported by JSPS KAKENHI (15H05707, 18K19793, and 18H03201), Japan Digital Design, and JST CREST.

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

## A  PROOF OF THEOREM 1

As we wrote in the main article, it is enough to show the following lemmas (definition of miscellaneous variables are as in Section 3.2). Remember that $\lambda = \sup_{n \in [N-M]} |\lambda_n|$ and $s_{lh}$ is the maximum singular value of $W_{lh}$

**Lemma 1.** *For any $X \in \mathbb{R}^{N \times C}$, we have $d_{\mathcal{M}}(PX) \leq \lambda d_{\mathcal{M}}(X)$.*

**Lemma 2.** *For any $X \in \mathbb{R}^{N \times C}$, we have $d_{\mathcal{M}}(XW_{lh}) \leq s_{lh} d_{\mathcal{M}}(X)$.*

**Lemma 3.** *For any $X \in \mathbb{R}^{N \times C}$, we have $d_{\mathcal{M}}(\sigma(X)) \leq d_{\mathcal{M}}(X)$.*

*Proof of Lemma 1.* Since $P$ is a symmetric linear operator on $U^\perp$, we can choose the orthonormal basis $(e_m)_{m=M+1,\ldots,N}$ of $U^\perp$ consisting of the eigenvalue of $P|_{U^\perp}$. Let $\lambda_m$ be the eigenvalue of $P$ to which $e_m$ is associated ($m = M + 1, \ldots, N$). Note that since the operator norm of $P|_{U^\perp}$ is $\lambda$, we have $|\lambda_m| \leq \lambda$ for all $m = M + 1, \ldots, N$. Since $(e_m)_{m \in [N]}$ forms the orthonormal basis of

$\mathbb{R}^N$, we can uniquely write $X \in \mathbb{R}^{N \times C}$ as $X = \sum_{m=1}^N e_m \otimes w_m$ for some $w_m \in \mathbb{R}^C$. Then, we have $d_{\mathcal{M}}^2(X) = \sum_{m=M+1}^N \|w_m\|^2$ where $\|\cdot\|$ is the 2-norm of a vector. On the other hand, we have

$$
\begin{aligned}
PX &= \sum_{m=1}^N Pe_m \otimes w_m \\
&= \sum_{m=1}^M Pe_m \otimes w_m + \sum_{m=M+1}^N Pe_m \otimes w_m \\
&= \sum_{m=1}^M Pe_m \otimes w_m + \sum_{m=M+1}^N e_m \otimes (\lambda_m w_m)
\end{aligned}
$$

Since $U$ is invariant under $P$, for any $m \in [M]$, we can write $Pe_m$ as a linear combination of $e_n (n \in [M])$. Therefore, we have $d_{\mathcal{M}}^2(PX) = \sum_{m=M+1}^N \|\lambda_m w_m\|^2$. Then, we obtain the desired inequality as follows:

$$
\begin{aligned}
d_{\mathcal{M}}^2(PX) &= \sum_{m=M+1}^N \|\lambda_m w_m\|^2 \\
&\le \lambda^2 \sum_{m=M+1}^N \|w_m\|^2 \\
&\le \lambda^2 \sum_{m=M+1}^N \|w_m\|^2 \\
&= \lambda^2 d_{\mathcal{M}}^2(X).
\end{aligned}
$$

$\square$

*Proof of Lemma 2.* Using the same decomposition of $X$ as the proof in Lemma 1, we have

$$
\begin{aligned}
XW_{lh} &= \sum_{m=1}^N e_m \otimes (W_{lh}^\top w_m) \\
&= \sum_{m=1}^M e_m \otimes (W_{lh}^\top w_m) + \sum_{m=M+1}^N e_m \otimes (W_{lh}^\top w_m).
\end{aligned}
$$

Therefore, we have

$$
\begin{aligned}
d_{\mathcal{M}}^2(XW_{lh}) &= \sum_{m=M+1}^N \|W_{lh}^\top w_m\|^2 \\
&\le s_{lh}^2 \sum_{m=M+1}^N \|w_m\|^2 \\
&= s_{lh}^2 d_{\mathcal{M}}^2(X).
\end{aligned}
$$

$\square$

*Proof of Lemma 3.* We choose $(e_m)_{m=N-M+1,\ldots,N}$ as in the proof of Lemma 1. We denote $X = (X_{nc})_{n \in [N], c \in [C]}$ and $e_n = (e_{mn})_{m \in [N]}$, respectively. Let $(e'_c)_{c \in [C]}$ be the standard basis of $\mathbb{R}^C$. Then, $(e_n \otimes e'_c)_{n \in [N], c \in [C]}$ is the orthonormal basis of $\mathbb{R}^{N \times C}$, endowed with the standard inner product as a Euclid space. Therefore, we can decompose $X$ as $X = \sum_{n=1}^N \sum_{c=1}^C a_{nc} e_n \otimes e'_c$ where $a_{nc} = \langle X, e_n \otimes e'_c \rangle = \sum_{m=1}^N X_{mc} e_{mn}$. Then, we have $d_{\mathcal{M}}^2(X) = \sum_{n=M+1}^N \|\sum_{c=1}^C a_{nc} e'_c\|^2$,

which is further transformed as

$$
\begin{aligned}
d_{\mathcal{M}}^2(X) &= \sum_{n=M+1}^{N} \left\| \sum_{c=1}^{C} a_{nc} e_c' \right\|^2 \\
&= \sum_{n=M+1}^{N} \sum_{c=1}^{C} a_{nc}^2 \\
&= \sum_{c=1}^{C} \left( \sum_{n=1}^{N} a_{nc}^2 - \sum_{n=1}^{M} a_{nc}^2 \right) \\
&= \sum_{c=1}^{C} \left( \|X_{\cdot c}\|^2 - \sum_{n=1}^{M} \langle X_{\cdot c}, e_n \rangle^2 \right),
\end{aligned}
$$

where $X_{\cdot c}$ is the $c$-th column vector of $X$. Similarly, we have

$$
d_{\mathcal{M}}^2(\sigma(X)) = \sum_{c=1}^{C} \left( \|X_{\cdot c}^+\|^2 - \sum_{n=1}^{M} \langle X_{\cdot c}^+, e_n \rangle^2 \right),
$$

where we denote $\sigma(X) = (X_{nc}^+)_{n \in [N], c \in [C]}$ in shorthand. Therefore, the inequality follow from the following lemma. □

**Lemma 4.** *Let $x \in \mathbb{R}^N$ and $v_1, \ldots, v_M \in \mathbb{R}^N$ be orthonormal vectors (i.e., $\langle v_m, v_n \rangle = \delta_{mn}$) satisfying $v_m \geq 0$ for all $m \in [M]$. Then, we have $\|x\|^2 - \sum_{m=1}^{M} \langle x, v_m \rangle^2 \geq \|x^+\|^2 - \sum_{m=1}^{M} \langle x^+, v_m \rangle^2$ where $x^+ := \max(x, 0)$ for $x \in \mathbb{R}$.*

*Proof.* The value $\|y\|^2 - \sum_{m=1}^{M} \langle y, u_m \rangle^2$ is invariant under simultaneous coordinate permutation of $y$ and $u_m$'s. Therefore, we can assume without loss of generality that the coordinate of $x$ are sorted: $x_1 \leq \ldots \leq x_L < 0 \leq x_{L+1} \leq \cdots \leq x_N$ for some $L \leq N$. Then, we have

$$
\|x\|^2 - \|x^+\|^2 = \sum_{n=1}^{L} x_n^2. \tag{1}
$$

When $L = 0$, the sum in the right hand side is treated as 0. On the other hand, writing as $v_m = (v_{nm})_{n \in [N]}$, direct calculation shows

$$
\sum_{m=1}^{M} \langle x, v_m \rangle^2 - \langle x^+, v_m \rangle^2 = \sum_{m=1}^{M} \left( \left( \sum_{n=1}^{L} x_n v_{nm} \right)^2 - 2 \sum_{n=1}^{L} \sum_{l=L+1}^{N} x_n x_l v_{nm} v_{lm} \right). \tag{2}
$$

Let $I_m := \{ n \in [N] \mid v_{nm} > 0 \}$ be the support of $v_m$ for $m \in [M]$. We note that if $m \neq m' \in [M]$, we have $I_m \cap I_{m'} = \emptyset$ since if there existed $n \in I_m \cap I_{m'}$, we have

$$
0 = \langle v_m, v_{m'} \rangle \geq v_{nm} v_{nm'} > 0,
$$

which is contradictory. Therefore,

$$
\begin{aligned}
\sum_{m=1}^{N} \left( \sum_{n=1}^{L} x_n v_{nm} \right)^2 &= \sum_{m=1}^{N} \left( \sum_{n \in I_m \cap [L]} x_n v_{nm} \right)^2 \\
&\leq \sum_{m=1}^{N} \left( \sum_{n \in I_m \cap [L]} x_n^2 \right) \left( \sum_{n \in I_m \cap [L]} v_{nm}^2 \right) \quad (\because \text{Cauchy–Schwarz inequality}) \\
&\leq \sum_{m=1}^{N} \left( \sum_{n \in I_m \cap [L]} x_n^2 \right) \quad (\because \|v_m\|^2 = 1) \\
&\leq \sum_{n=1}^{L} x_n^2. \tag{3}
\end{aligned}
$$

We used the fact that $I_m$'s are disjoint and $v_{nm} = 0$ if $n \notin \cup_m I_m$ in the first equality above. Further, we have $x_n x_l v_{nm} v_{lm} \leq 0$ for $1 \leq n \leq L$ and $L + 1 \leq l \leq N$ by the definition of $L$ and non-negativity of $v_m$. By combining (1), (2), and (3), we have

$$\sum_{m=1}^{M} \langle x, v_m \rangle^2 - \langle x^+, v_m \rangle^2 \leq \sum_{n=1}^{L} x_n^2 = \|x\|^2 - \|x^+\|^2.$$

$\square$

*Proof of Theorem 1.* By Lemma 1, 2, and 3, we have

$$
\begin{aligned}
d_{\mathcal{M}}(f_l(X)) &= d_{\mathcal{M}}(\underbrace{\sigma(\cdots \sigma(\sigma(PX)W_{l1})W_{l2} \cdots W_{lH_l})}_{H \text{ times}}) \\
&\leq d_{\mathcal{M}}(\underbrace{\sigma(\cdots \sigma(\sigma(PX)W_{l1})W_{l2} \cdots)}_{H-1 \text{ times}} W_{lH_l}) \\
&\leq s_{lH_l-1} d_{\mathcal{M}}(\underbrace{\sigma(\cdots \sigma(\sigma(PX)W_{l1})W_{l2} \cdots)}_{H-1 \text{ times}} W_{lH_l-1})) \\
&\cdots \\
&\leq \left( \prod_{h=1}^{H_l} s_{lh} \right) d_{\mathcal{M}}(PX) \\
&\leq s_l d_{\mathcal{M}}(PX) \\
&\leq s_l \lambda d_{\mathcal{M}}(X).
\end{aligned}
$$

$\square$

# B  PROOF OF PROPOSITION 1

*Proof.* Let $\tilde{\mu}_1 \leq \cdots \leq \tilde{\mu}_N$ be the eigenvalue of the augmented normalized Laplacian $\tilde{\Delta}$, sorted in ascending order. Since $P = I_N - \tilde{\Delta}$, it is enough to show $\tilde{\mu}_1 = \cdots = \tilde{\mu}_M = 0$, $\tilde{\mu}_{M+1} > 0$, and $\tilde{\mu}_N < 2$. For the first two, the statements are equivalent to that $\tilde{\Delta}$ is positive semi-definite and that the multiplicity of the eigenvalue 0 is same as the number of connected components [6]. This is well-known for Laplacian or its normalized version (see, e.g., Chung & Graham (1997)) and the proof for $\tilde{\Delta}$ is similar. By direct calculation, we have

$$x^\top \tilde{\Delta} x = \frac{1}{2} \sum_{i,j=1}^{N} a_{ij} \left( \frac{x_i}{\sqrt{d_i + 1}} - \frac{x_j}{\sqrt{d_j + 1}} \right)^2$$

for any $x = [x_1 \ \cdots \ x_N]^\top \in \mathbb{R}^N$. Therefore, $\tilde{\Delta}$ is positive semi-definite and hence $\tilde{\mu}_1 \geq 0$.

Suppose temporally that $G$ is connected. If $x \in \mathbb{R}^N$ is an eigenvector associated to 0, then, by the aforementioned calculation, $\frac{x_i}{\sqrt{d_i+1}}$ and $\frac{x_j}{\sqrt{d_j+1}}$ must be same for all pairs $(i,j)$ such that $a_{ij} > 0$. However, since $G$ is connected, $\frac{x_i}{\sqrt{d_i+1}}$ must be same value for all $i \in [N]$. That means the multiplicity of the eigenvalue 0 is 1 and any eigenvector associated to 0 must be proportional to $\tilde{D}^{\frac{1}{2}} \mathbf{1}$. Now, suppose $G$ has $M$ connected components $V_1, \ldots, V_M$. Let $\tilde{\Delta}_m$ be the augmented normalized Laplacians corresponding to each connected component $V_m$ for $m \in [M]$. By the aforementioned discussion, $\tilde{\Delta}_m$ has the eigenvalue 0 with multiplicity 1. Since $\tilde{\Delta}$ is the direct sum of $\tilde{\Delta}'_m s$, the eigenvalue of $\tilde{\Delta}$ is the union of those for $\tilde{\Delta}_m$'s. Therefore, $\tilde{\Delta}$ has the eigenvalue 0 with multiplicity $M$ and $e_m = \tilde{D}^{\frac{1}{2}} \mathbf{1}_m$'s are the orthogonal basis of the eigenspace.

---

[6]The former statement is identical to Lemma 1 and latter one is the extension of Lemma 2 of Wu et al. (2019a).

Finally, we prove $\tilde{\mu}_N < 2$. Let $\mu_N$ be the largest eigenvalue of the normalized Laplacian $\Delta = D^{-\frac{1}{2}}(D-A)D^{-\frac{1}{2}}$, where $D^{-\frac{1}{2}} \in \mathbb{R}^{N \times N}$ is the diagonal matrix defined by

$$D_{ii}^{-\frac{1}{2}} = \begin{cases} \deg(i)^{-\frac{1}{2}} & (\text{if } \deg(i) \neq 0) \\ 0 & (\text{if } \deg(i) = 0) \end{cases}.$$

Note that $D^{-\frac{1}{2}}D^{\frac{1}{2}}$ nor $D^{\frac{1}{2}}D^{-\frac{1}{2}}$ are not equal to the identity matrix $I_N$ in general. However, we have

$$L = D^{\frac{1}{2}}D^{-\frac{1}{2}}LD^{-\frac{1}{2}}D^{\frac{1}{2}} \tag{4}$$

where $L = D - A$ is the (unnormalized) Laplacian. Therefore, we have

$$\begin{aligned}
\tilde{\mu}_N &= \max_{x \neq 0} \frac{x^\top \tilde{\Delta} x}{\|x\|} \\
&= \max_{x \neq 0} \frac{x^\top \tilde{D}^{-\frac{1}{2}} L \tilde{D}^{-\frac{1}{2}} x}{\|x\|} \\
&= \max_{x \neq 0} \frac{x^\top \tilde{D}^{-\frac{1}{2}} D^{\frac{1}{2}} D^{-\frac{1}{2}} L D^{-\frac{1}{2}} D^{\frac{1}{2}} \tilde{D}^{-\frac{1}{2}} x}{\|x\|} \quad (\because (4)) \\
&= \max_{x \neq 0} \frac{(D^{\frac{1}{2}}\tilde{D}^{-\frac{1}{2}}x)^\top \Delta (D^{\frac{1}{2}}\tilde{D}^{-\frac{1}{2}}x)}{\|x\|} \\
&= \max_{\substack{x \neq 0 \\ D^{\frac{1}{2}}\tilde{D}^{-\frac{1}{2}}x \neq 0}} \frac{(D^{\frac{1}{2}}\tilde{D}^{-\frac{1}{2}}x)^\top \Delta (D^{\frac{1}{2}}\tilde{D}^{-\frac{1}{2}}x)}{\|x\|} \\
&= \max_{\substack{x \neq 0 \\ D^{\frac{1}{2}}\tilde{D}^{-\frac{1}{2}}x \neq 0}} \frac{(D^{\frac{1}{2}}\tilde{D}^{-\frac{1}{2}}x)^\top \Delta (D^{\frac{1}{2}}\tilde{D}^{-\frac{1}{2}}x)}{\|D^{\frac{1}{2}}\tilde{D}^{-\frac{1}{2}}x\|} \frac{\|D^{\frac{1}{2}}\tilde{D}^{-\frac{1}{2}}x\|}{\|x\|} \\
&\leq \max_{\substack{x \neq 0 \\ D^{\frac{1}{2}}\tilde{D}^{-\frac{1}{2}}x \neq 0}} \frac{(D^{\frac{1}{2}}\tilde{D}^{-\frac{1}{2}}x)^\top \Delta (D^{\frac{1}{2}}\tilde{D}^{-\frac{1}{2}}x)}{\|D^{\frac{1}{2}}\tilde{D}^{-\frac{1}{2}}x\|} \max_{\substack{x \neq 0 \\ D^{\frac{1}{2}}\tilde{D}^{-\frac{1}{2}}x \neq 0}} \frac{\|D^{\frac{1}{2}}\tilde{D}^{-\frac{1}{2}}x\|}{\|x\|} \\
&\leq \max_{y \neq 0} \frac{y^\top \Delta y}{\|y\|} \max_{x \neq 0} \frac{\|D^{\frac{1}{2}}\tilde{D}^{-\frac{1}{2}}x\|}{\|x\|} \\
&= \mu_N \max_{n \in [N]} \left( \frac{\deg(i)}{\deg(i)+1} \right)^{\frac{1}{2}} \\
&\leq \mu_N.
\end{aligned}$$

Therefore, we have $\tilde{\mu}_N \leq \mu_N$[7]. Since $\max_{i \in [N]} \left( \frac{\deg(i)}{\deg(i)+1} \right)^{\frac{1}{2}} < 1$, the equality $\tilde{\mu}_N = \mu_N$ holds if and only if $\mu_N = 0$, that is, $G$ has $N$ connected components. On the other hand, it is known that $\mu_N \leq 2$ and the equality holds if and only if $G$ has non-trivial bipartite graph as a connected component (see, e.g., Chung & Graham (1997)). Therefore, $\tilde{\mu}_N = \mu_N$ and $\mu_N = 2$ does not hold simultaneously and we obtain $\mu_N < 2$. □

# C COUNTEREXAMPLE OF PREVIOUS STUDY ON OVER-SMOOTHING FOR NON-LINEAR GNNS

We restate Theorem 1 of the preprint (version2) of Luan et al. (2019)[8]. Let $G$ be a simple undirected graph with $N$ nodes and $k$ connected components such that it does not have a bipartite component. Let $L = \tilde{D}^{-1/2}\tilde{A}\tilde{D}^{-1/2} \in \mathbb{R}^{N \times N}$ be the augmented normalized Laplacian of $G$. Let $F \in \mathbb{N}_+$ and

---

[7]Theorem 1 of Wu et al. (2019a) showed that this inequality strictly holds when $G$ is simple and connected. We do not require this assumption.

[8]https://arxiv.org/abs/1906.02174v2

$W_n \in \mathbb{R}^{F \times F}$ be the weight of the $n$-th layer for $n \in \mathbb{N}_+$. For the input $X \in \mathbb{R}^{N \times F}$, we define the output $Y_n \in \mathbb{R}^{N \times F}$ of the $n$-th layer of a GCN by $Y_n = \sigma(L \cdots \sigma(LXW_0) \cdots W_n)$ where $\sigma$ is the ReLU function. We assume the input $X$ is drawn from a continuous distribution on $\mathbb{R}^{N \times F}$. Then, the theorem claims that we have $\lim_{n \to \infty} \text{rank}(Y_n) = k$ almost surely with respect to the distribution of $X$.

We construct a conterexample. Consider a graph $G$ consisting of $N = 4$ nodes whose adjacency matrix is

$$A = \begin{bmatrix} 1 & 1 & 1 & 1 \\ 1 & 1 & 1 & 0 \\ 1 & 1 & 1 & 0 \\ 1 & 0 & 0 & 1 \end{bmatrix}.$$

Note that $G$ is connected (i.e., $k = 1$) and is not bipartite. We make a GCN with $F = 3$ channels and whose weight matrices are $W_n = I_3$ (the identity matrix of size 3) for all $n \in \mathbb{N}$. For the distribution of the input $X$, we consider an absolutely continuous distribution with respect to the Lebesgue measure on $\mathbb{R}^{4 \times 3}$ such that $P(X \geq 0) > 0$ (here, $X \geq 0$ means the element-wise comparison). For example, the standard Gaussian distribution satisfies the condition.

Since $L \geq 0$, we have $Y_n = L^n X$ if $X \geq 0$. Let $L = P^\top \Lambda P$ be the diagonalization of $L$ where $P \in O(4)$ is an orthogonal matrix of size 4. Since $\text{rank}(L) = 3$, we have $\text{rank}(\Lambda^n) = 3$ for any $n$ (we can assume that $\Lambda_{44} = 0$ without loss of generality). Therefore, under the condition $X \geq 0$, we have

$$\text{rank}(Y_n) = 3 \iff \text{rank}(P^\top \Lambda^n P X) = 3$$
$$\iff X \in \{P^{-1} \begin{bmatrix} B & v \end{bmatrix}^\top \mid B \in \mathbb{R}^{3 \times 3} \text{ is invertible}, v \in \mathbb{R}^3\}.$$

Note that the last condition is independent of $n$. Since the set of invertible matrices is dense in the set of all matrices of the same size (with respect to the standard topology of the Euclidean space), we have $P(\{\text{rank}(Y_n) = 3 \text{ for all } n \in \mathbb{N}\}) > 0$. Therefore, we have $\lim_{n \to \infty} \text{rank}(Y_n) = 3$ with a non-zero probability. $\qquad\square$

## D  PROOF OF THEOREM 3

We follow the proof of Theorem 2 of Chung & Radcliffe (2011). The idea is to relate the spectral distribution of the normalized Laplacian with that of its expected version. Since we can compute the latter one explicitly for the Erdős-Rényi graph, we can derive the convergence of spectra. We employ this technique and derive similar conclusion for the augmented normalized Laplacian.

First, we consider genral random graphs not restricted to Erdős-Rényi graphs. Let $N \in \mathbb{N}_+$, and $P = (p_{ij})_{i,j \in [N]}$ be a non-negative symmetric matrix (meaning that $p_{ij} \geq 0$ for any $i, j \in [N]$). Let $G$ be an undirected random graph with $N$ nodes such that an edge between $i$ and $j$ is independently present with probability $p_{ij}$. Let $A$ and $D$ be the adjacency and the degree matrices of $G$, respectively (that is, $A_{ij} \sim \text{Ber}(p_{ij})$, i.i.d.). Define the expected node degree of node $i$ by $t_i := \sum_{j=1}^{N} p_{ij}$. Let $\tilde{A} := A + I_N$, $\tilde{D} := D + I_N$ and define $\bar{A} := \mathbb{E}[\tilde{A}] = P + I_N$ and $\bar{D} := \mathbb{E}[\tilde{D}] = \text{diag}(t_1, \ldots, t_N) + I_N$ correspondingly. We define the augmented normalized Laplacian $\tilde{\Delta}$ of $G$ by $\tilde{\Delta} := I_N - \tilde{D}^{-\frac{1}{2}} \tilde{A} \tilde{D}^{-\frac{1}{2}}$ and its expected version by $\bar{\Delta} := I_N - \bar{D}^{-\frac{1}{2}} \bar{A} \bar{D}^{-\frac{1}{2}}$ [9]. For a symmetric matrix $X \in \mathbb{R}^N$, we define its eigenvalues, sorted in ascending order by $\lambda_1(X) \leq \cdots \leq \lambda_N(X)$ and its operator norm by $\|X\| = \max_{n \in [N]} |\lambda_n(X)|$.

**Lemma 5** (Ref. Chung & Radcliffe (2011) Theorem 2). *Let $\delta := \min_{n \in [N]} t_n$ be the minimum expected degree of $G$. Set $k(\varepsilon) := 3(1 + \log(4/\varepsilon))$. Then, for any $\varepsilon > 0$, if $\delta + 1 > k(\varepsilon) \log N$, we have*

$$\max_{n \in [N]} \left| \lambda_n(\tilde{\Delta}) - \lambda_n(\bar{\Delta}) \right| \leq 4\sqrt{\frac{3 \log(4N/\varepsilon)}{\delta + 1}}$$

*with probability at least $1 - \varepsilon$.*

---

[9]Note that $\mathbb{E}[\tilde{\Delta}] \neq \bar{\Delta}$ in general due to the dependence between $\tilde{A}$ and $\tilde{D}$.

*Proof.* By Weyl's theorem, we have $\max_{n \in [N]} \left| \lambda_n(\tilde{\Delta}) - \lambda_n(\bar{\Delta}) \right| \leq \|\tilde{\Delta} - \bar{\Delta}\|$. Therefore, it is enough to bound $\|\tilde{\Delta} - \bar{\Delta}\|$. Let $C := I_N - \bar{D}^{-\frac{1}{2}} \tilde{A} \bar{D}$. By the triangular inequality, we have $\|\tilde{\Delta} - \bar{\Delta}\| \leq \|\tilde{\Delta} - C\| + \|C - \bar{\Delta}\|$. We will bound these terms respectively.

First, we bound $\|C - \bar{\Delta}\|$. Direct calculation shows $C - \bar{\Delta} = -\bar{D}^{-\frac{1}{2}}(A - P)\bar{D}^{-\frac{1}{2}}$. Let $E^{ij} \in \mathbb{R}^{N \times N}$ be a matrix defined by

$$(E^{ij})_{kl} = \begin{cases} 1 & \text{if } (i = k \text{ and } i = l) \text{ or } (i = l \text{ and } j = k), \\ 0 & \text{otherwise.} \end{cases}$$

We define the random variable $Y_{ij}$ by

$$Y_{ij} := \frac{A_{ij} - p_{ij}}{\sqrt{t_i + 1}\sqrt{t_j + 1}} E^{ij}.$$

Then, $Y_{ij}$'s are independent and we have $C - \bar{\Delta} = \sum_{i,j=1}^{N} Y_{ij}$. To apply Theorem 5 of Chung & Radcliffe (2011) to $Y_{ij}$'s, we bound $\|Y_{ij} - \mathbb{E}[Y_{ij}]\|$ and $\|\sum_{i,j=1}^{N} \mathbb{E}[Y_{ij}^2]\|$. First, we have

$$\|Y_{ij} - \mathbb{E}[Y_{ij}]\| = \|Y_{ij}\| \leq \frac{\|E^{ij}\|}{\sqrt{t_i + 1}\sqrt{t_j + 1}} \leq (\delta + 1)^{-1}.$$

Since

$$\mathbb{E}[Y_{ij}^2] = \frac{p_{ij} - p_{ij}^2}{(t_i + 1)(t_j + 1)} \begin{cases} E^{ii} + E^{jj} & (\text{if } i \neq j), \\ E^{ii} & (\text{if } i = j), \end{cases}$$

we have

$$\left\| \sum_{i,j=1}^{N} \mathbb{E}[Y_{ij}^2] \right\| = \left\| \sum_{i,j=1}^{N} \frac{p_{ij} - p_{ij}^2}{(t_i + 1)(t_j + 1)} E^{ii} \right\|$$

$$= \max_{i \in [N]} \left( \sum_{j=1}^{N} \frac{p_{ij} - p_{ij}^2}{(t_i + 1)(t_j + 1)} \right)$$

$$\leq \max_{i \in [N]} \left( \sum_{j=1}^{N} \frac{p_{ij}}{(t_i + 1)(t_j + 1)} \right)$$

$$\leq (\delta + 1)^{-1}.$$

By letting $a \leftarrow \sqrt{\frac{3\log(4N/\varepsilon)}{\delta + 1}}$, $M \leftarrow (\delta + 1)^{-1}$, $v^2 \leftarrow (\delta + 1)^{-1}$ and applying Theorem 5 of Chung & Radcliffe (2011), we have

$$\Pr(\|C - \bar{\Delta}\| > a) \leq 2N \exp\left( -\frac{a^2}{2(\delta + 1)^{-1} + 2(\delta + 1)^{-1} a/3} \right)$$

$$\leq 2N \exp\left( -\frac{3\log(4N/\varepsilon)}{2(1 + a/3)} \right).$$

By the definition of $k(\varepsilon)$, we have $a < 1$ if $\delta + 1 > k(\varepsilon) \log n$. For such $\delta$, we have

$$\Pr(\|C - \bar{\Delta}\| > a) \leq 2N \exp\left( -\frac{3\log(4N/\varepsilon)}{2(1 + a/3)} \right)$$

$$\leq 2N \exp\left( -\log(4N/\varepsilon) \right) \quad (\because a < 1)$$

$$= \frac{\varepsilon}{2}. \tag{5}$$

Next, we bound $\|\tilde{\Delta} - C\|$. First, since $a < 1$, by Chernoff bound (see, e.g. Angluin & Valiant (1979); Hagerup & Rüb (1990))), we have

$$
\begin{aligned}
\Pr(|d_i - t_i| > a(t_i + 1)) &\leq 2\exp\left(-\frac{a^2(t_i + 1)}{3}\right) \\
&\leq 2\exp\left(-\frac{a^2(\delta + 1)}{3}\right) \\
&= \frac{\varepsilon}{2N}.
\end{aligned}
$$

Therefore, if $|d_i - t_i| \leq a(t_i + 1)$, then we have

$$
\begin{aligned}
\left|\sqrt{\frac{d_i + 1}{t_i + 1}} - 1\right| &\leq \left|\frac{d_i + 1}{t_i + 1} - 1\right| \quad (\because |\sqrt{x} - 1| \leq |x - 1| \text{ for } x \geq 0) \\
&= \left|\frac{d_i - t_i}{t_i + 1}\right| \\
&\leq a.
\end{aligned}
$$

Therefore, by union bound, we have

$$
\|\bar{D}^{-\frac{1}{2}}\tilde{D}^{\frac{1}{2}} - I_N\| = \max_{i \in [N]}\left|\sqrt{\frac{d_i + 1}{t_i + 1}} - 1\right| \leq a
$$

with probability at least $1 - \varepsilon/2$. Further, since the eigenvalue of the augmented normalized Laplacian is in $[0, 2]$ by the proof of Proposition 1, we have $\|I_N - \tilde{\Delta}\| \leq 1$. By combining them, we have

$$
\begin{aligned}
\|\tilde{\Delta} - C\| &= \|(\bar{D}^{-\frac{1}{2}}\tilde{D}^{\frac{1}{2}} - I_N)(I_N - \tilde{\Delta})\tilde{D}^{\frac{1}{2}}\bar{D}^{-\frac{1}{2}} + (I_N - \tilde{\Delta})(I - \tilde{D}^{\frac{1}{2}}\bar{D}^{-\frac{1}{2}})\| \\
&\leq \|(\bar{D}^{-\frac{1}{2}}\tilde{D}^{\frac{1}{2}} - I_N)\|\|\tilde{D}^{\frac{1}{2}}\bar{D}^{-\frac{1}{2}}\| + \|I - \tilde{D}^{\frac{1}{2}}\bar{D}^{-\frac{1}{2}}\| \\
&\leq a(a + 1) + a. \tag{6}
\end{aligned}
$$

From (5) and (6), we have

$$
\begin{aligned}
\|\tilde{\Delta} - \bar{\Delta}\| &\leq \|\tilde{\Delta} - C\| + \|C - \bar{\Delta}\| \\
&\leq a + a(a + 1) + a \\
&\leq a^2 + 3a \\
&\leq 4a \quad (\because a < 1)
\end{aligned}
$$

with probability at least $1 - \varepsilon$ by union bound. $\qquad\square$

Let $N \in \mathbb{N}_+$ and $p > 0$. In the case of the Erdős-Rényi graph $G_{N,p}$, we should set $P = p(J_N - I_N)$ where $J_N \in \mathbb{R}^{N \times N}$ are the all-one matrix. Then, we have $\bar{A} = pJ_N + (1 - p)I_N$, $\bar{D} = (Np - p + 1)I_N$, and $\bar{\Delta} = \frac{p}{Np - p + 1}(NI_N - J_N)$. Since the eigenvalue of $J_N$ is $N$ (with multiplicity 1) and 0 (with multiplicity $N - 1$), the eigenvalue of $\bar{\Delta}$ is 0 (with multiplicity 1) and $\frac{Np}{Np - p + 1}$ (with multiplicity $N - 1$). For $G_{N,p}$, $\delta$ is the expected average degree $(N - 1)p$. Hence, we have the following lemma from Lemma 5:

**Lemma 6.** *Let $\tilde{\Delta}$ be its augmented normalized Laplacian of the Erdős-Rényi graph $G_{N,p}$. For any $\varepsilon > 0$, if $\frac{Np - p + 1}{\log N} > k(\varepsilon) := 3(1 + \log(4/\varepsilon))$, then, with probability at least $1 - \varepsilon$, we have*

$$
\max_{i=2,\ldots,N}\left|\lambda_i(\tilde{\Delta}) - \frac{Np}{Np - p + 1}\right| \leq 4\sqrt{\frac{3\log(4N/\varepsilon)}{Np - p + 1}}.
$$

**Corollary 4.** *Consider GCN on $G_{N,p}$. Let $W_l$ be the weight of the $l$-th layer of GCN and $s_l$ be the maximum singular value of $W_l$ for $l \in \mathbb{N}_+$. Set $s := \sup_{l \in \mathbb{N}_+}$. Let $\varepsilon > 0$. We define $k(\varepsilon) := 3(1 + \log(4/\varepsilon))$ and $l(N, p, \varepsilon) = \frac{1 - p}{Np - p + 1} + 4\sqrt{\frac{3\log(4N/\varepsilon)}{Np - p + 1}}$. If $\frac{Np - p + 1}{\log N} > k(\varepsilon)$ and $s \leq l(N, \varepsilon)^{-1}$, then, GCN on $G_{N,p}$ satisfies the assumption of Theorem 2 with probability at least $1 - \varepsilon$.*

*Proof of Theorem 3.* Since $\frac{\log N}{Np} = o(1)$, for fixed $\varepsilon$, we have

$$\frac{Np - p + 1}{\log N} > \frac{Np}{\log N} > k(\varepsilon)$$

for sufficiently large $N$. Further, $Np \to \infty$ as $N \to \infty$ when $\frac{\log N}{Np} = o(1)$. Therefore, we have

$$\frac{(1-p)^2}{Np - p + 1} \leq \frac{1}{Np} \leq (7 - 4\sqrt{3})^2 \log\left(\frac{4N}{\varepsilon}\right)$$

for sufficiently large $N$. Hence.

$$\frac{1-p}{Np - p + 1} \leq (7 - 4\sqrt{3}) \sqrt{\frac{\log(4N/\varepsilon)}{Np - p + 1}}.$$

Therefore, we have $l(N, p, \varepsilon) \leq 7\sqrt{\frac{\log(4N/\varepsilon)}{Np-p+1}}$. Therefore, if $s \leq \frac{1}{7}\sqrt{\frac{Np-p+1}{\log(4N/\varepsilon)}}$, then we have $s \leq l(N, p, \varepsilon)^{-1}$. $\qquad\square$

# E  MISCELLANEOUS PROPOSITIONS

## E.1  INVARIANCE OF ORTHOGONAL COMPLEMENT SPACE

**Proposition 2.** *Let $P \in \mathbb{R}^{N \times N}$ be a symmetric matrix, treated as a linear operator $P : \mathbb{R}^N \to \mathbb{R}^N$. If a subspace $U \subset \mathbb{R}^N$ is invariant under $P$ (i.e., if $u \in U$, then $Pu \in U$), then, $U^\perp$ is invariant under $P$, too.*

*Proof.* For any $u \in U^\perp$ and $v \in U$, by symmetry of $P$, we have

$$\langle Pu, v \rangle = (Pu)^\top v = u^\top P^\top v = u^\top Pv = \langle u, Pv \rangle.$$

Since $U$ is an invariant space of $P$, we have $Pv \in U$. Hence, we have $\langle u, Pv \rangle = 0$ because $u \in U^\perp$. We obtain $Pu \in U^\perp$ by the definition of $U^\perp$. $\qquad\square$

## E.2  CONVERGENCE TO TRIVIAL FIXED POINT

Let $P \in \mathbb{R}^{N \times N}$ be a symmetric matrix, $W_l \in \mathbb{R}^{C \times C}$, $s_l$ be the maximum singular value of $W_l$ for $l \in \mathbb{N}_+$. We define $f_l : \mathbb{R}^{N \times C} \to \mathbb{R}^{N \times C}$ by $f_l(X) := \sigma(PXW_l)$ where $\sigma$ is the element-wise ReLU function.

**Proposition 3.** *Suppose further that the operator norm of $P$ is no larger than $\lambda$, then we have $\|f_l(X)\|_{\mathrm{F}} \leq s_l \lambda \|X\|_{\mathrm{F}}$ for any $l \in \mathbb{N}_+$. In particular, let $s := \sup_{l \in \mathbb{N}_+} s_l$. If $s\lambda < 1$, then, $X_l$ exponentially approaches $0$ as $l \to \infty$.*

*Proof.* Since $\lambda$ is the operator norm of $P|_{U^\perp}$, the assumption implies that the operator norm of $P$ itself is no larger than $\lambda$. Therefore, we have $\|PXW_l\|_{\mathrm{F}} \leq \lambda\|XW_l\|_{\mathrm{F}} \leq s_l\lambda\|X\|_{\mathrm{F}}$. On the other hand, since $\sigma(x)^2 \leq x^2$ for any $x \in \mathbb{R}$, we have $\|\sigma(X)\|_{\mathrm{F}} \leq \|X\|_F$ for any $X \in \mathbb{R}^{N \times C}$. Combining the two, we have $\|f_l(X)\|_{\mathrm{F}} \leq \|PXW_l\|_{\mathrm{F}} \leq s_l\lambda\|X\|_{\mathrm{F}}$. $\qquad\square$

## E.3  STRICTNESS OF THEOREM 1

Theorem 1 implies that if $s\lambda \leq 1$, then, one-step transition $f_l$ does not increase the distance to $\mathcal{M}$. In this section, we first prove that this theorem is strict in the sense that, there exists a situation in which $s_l\lambda > 1$ holds and the distance $d_\mathcal{M}$ increases by one-step transition $f_l$ at some point $X$.

Set $N \leftarrow 2$, $C \leftarrow 1$, and $M \leftarrow 1$ in Section 3.2. For $\mu, \lambda > 0$, we set

$$P \leftarrow \begin{bmatrix} \mu & 0 \\ 0 & \lambda \end{bmatrix}, e \leftarrow \begin{bmatrix} 1 \\ 0 \end{bmatrix}, U \leftarrow \left\{ \begin{bmatrix} x \\ y \end{bmatrix} \mid y = 0 \right\}.$$

Then, by definition, we can check that the 3-tuple $(P, e, U)$ satisfies the Assumptions 1 and 2. Set $\mathcal{M} := U \otimes \mathbb{R} = U$ and choose $W \in \mathbb{R}$ so that $W > \lambda^{-1}$. Finally define $f : \mathbb{R}^{N \times C} \to \mathbb{R}^{N \times C}$ by $f(X) := \sigma(PXW)$ where $\sigma$ is the element-wise ReLU function.

**Proposition 4.** *We have $d_{\mathcal{M}}(f(X)) > d_{\mathcal{M}}(X)$ for any $X = [x_1 \quad x_2]^\top \in \mathbb{R}^2$ such that $x_2 > 0$.*

*Proof.* By definition, we have $d_{\mathcal{M}}(X) = |x_2|$. On the other hand, direct calculation shows that $f_l(X) = \left[(W\mu X_1)^+ \quad (W\lambda X_2)^+\right]^\top$ and $d_{\mathcal{M}}(f_l(X)) = (W\lambda X_2)^+$ where $x^+ := \max(x, 0)$ for $x \in \mathbb{R}$. Since $W > \lambda^{-1}$ and $x_2 > 0$, we have $d_{\mathcal{M}}(f_l(X)) > d_{\mathcal{M}}(X)$. $\qquad\square$

Next, we prove the non-strictness of Theorem 1 in the sense that there exists a situation in which $s_l\lambda > 1$ holds and the distance $d_{\mathcal{M}}$ uniformly decreases by $f_l$. Again, we set Set $N \leftarrow 2$, $C \leftarrow 1$, and $M \leftarrow 1$. Let $\lambda \in (1, 2)$ and set

$$P \leftarrow \frac{\lambda}{2} \begin{bmatrix} 1 & -1 \\ -1 & 1 \end{bmatrix}, e \leftarrow \frac{1}{\sqrt{2}} \begin{bmatrix} 1 \\ 1 \end{bmatrix}, U \leftarrow \left\{ \begin{bmatrix} x \\ y \end{bmatrix} \mid x = y \right\}$$

Then, we can directly show that 3-tuple $(P, e, U)$ satisfies the Assumptions 1 and 2. Set $W \leftarrow 1$.

**Proposition 5.** *We have $W\lambda > 1$ and $d_{\mathcal{M}}(f_l(X)) < d_{\mathcal{M}}(X)$ for all $X \in \mathbb{R}^2$.*

*Proof.* First, note that $e' := \frac{1}{\sqrt{2}}[1 \quad -1]^\top$ is the eigenvector of $P$ associated to $\lambda$: $Pe' = \lambda e'$. For $X = ae + be'$ $(a, b \; \text{¿} \; 0)$, the distance between $X$ and $\mathcal{M}$ is $d_{\mathcal{M}}(X) = |b|$. On the other hand, by direct computation, we have

$$f(X) = \sigma(PXW) = \begin{cases} \begin{bmatrix} 0 & \frac{\lambda b}{\sqrt{2}} \end{bmatrix}^\top & (\text{if } b \geq 0), \\ \begin{bmatrix} \frac{-\lambda b}{\sqrt{2}} & 0 \end{bmatrix}^\top & (\text{if } b < 0). \end{cases}$$

Therefore, the distance between $f(X)$ and $\mathcal{M}$ is $d_{\mathcal{M}}(f(X)) = \lambda|b|/2$. Since $\lambda < 2$, we have $d_{\mathcal{M}}(f(X)) < d_{\mathcal{M}}(X)$ for any $X \in \mathbb{R}^2$. $\qquad\square$

We have shown that the non-negativity of $e$ (Assumption 1) is *not* a redundant condition in Section 6.1.

# F  RELATION TO MARKOV PROCESS

It is known that any Markov process on finite states converges to a unique distribution (*equilibrium*) if it is irreducible and aperiodic (see, e.g., Norris (1998)). As we see in this section, this theorem is the special case of Corollary 3.

Let $S := \{1, \ldots, N\}$ be a finite discrete state space. Consider a Markov process on $S$ characterized by a symmetric transition matrix $P = (p_{ij})_{i,j\in[N]} \in \mathbb{R}^{N\times N}$ such that $P \geq 0$ and $P\mathbf{1} = \mathbf{1}$ where $\mathbf{1}$ is the all-one vector. We interpret $p_{ij}$ as the transition probability from a state $i$ to $j$. We associate $P$ with a graph $G_P = (V_P, E_P)$ by $V_P = [N]$ and $(i, j) \in E_P$ if and only if $p_{ij} > 0$. Since $P$ is symmetric, we can regard $G_P$ as an undirected graph. We assume $P$ is irreducible and aperiodic [10]. Perron – Frobenius' theorem (see, e.g., Meyer (2000)) implies that $P$ satisfy the assumption of Corollary 3 with $M = 1$.

**Proposition 6** (Perron – Frobenius). *Let the eigenvalues of $P$ be $\lambda_1 \leq \cdots \leq \lambda_N$. Then, we have $-1 < \lambda_1$, $\lambda_{N-1} < 1$, and $\lambda_N = 1$. Further, there exists unique vector $e \in \mathbb{R}^N$ such that $e \geq 0$, $\|e\| = 1$, and $e$ is the eigenvector for the eivenvalue 1.*

**Corollary 5.** *Let $\lambda := \max_{n=1,\ldots,N-1} |\lambda_n|(< 1)$ and $\mathcal{M} := \{e \otimes w \mid w \in \mathbb{R}^C\}$. If $s\lambda < 1$, then, for any initial value $X_1$, $X_l$ exponentially approaches $\mathcal{M}$ as $l \to \infty$.*

If we set $C = 1$ and $W_l = 1$ for all $l \in \mathbb{N}_+$, then, we can inductively show that $X_l \geq 0$ for any $l \geq 2$. Therefore, we can interpret $X_l$ as a measure on $S$. Suppose further that we take the initial value $X_1$ as $X_1 \geq 0$ and $X_1^\top \mathbf{1} = 1$ so that we can interpret $X_1$ as a probability distribution on $S$. Then, we can inductively show that $X_l \geq 0$, $X_l^\top \mathbf{1} = 1$ (i.e., $X_l$ is a probability distribution on

---

[10]A symmetric matrix $A$ is called *irreducible* if and only if $G_A$ is connected. We say a graph $G$ is *aperiodic* if the greatest common divisor of length of all loops in $G$ is 1. A symmetric matrix $A$ is aperiodic if the graph $G_A$ induced by $A$ is aperiodic.

$S$), and $X_{l+1} = \sigma(PX_lW_l) = PX_l$ for all $l \in \mathbb{N}_+$. In conclusion, the corollary is reduced to the fact that if a finite and discrete Markov process is irreducible and aperiodic, any initial probability distribution converges exponentially to an equbrilium. In addition, the the rate $\lambda$ corresponds to the *mixing time* of the Markov process.

## G   GCN DEFINED BY NORMALIZED LAPLACIAN

In Section 4, we defined $P$ using the augmented normalized Laplacian $\tilde{\Delta}$ by $P = I_N - \tilde{\Delta}$. We can alternatively use the usual normalized Laplacian $\Delta$ instead of the augmented one to define $P$ and want to apply the theory developed in Section 3.2. We write the normalized Laplacian version as $P_\Delta := I_N - \Delta$. The only obstacle is that the smallest eigenvalue $\lambda_1$ of $P_\Delta$ can be equal to $-1$, while that of $P$ is strictly larger than $-1$ (see, Proposition 1). This corresponds to that fact the largest eigenvalue of $\tilde{\Delta}$ is strictly smaller than 2, while that for $\Delta$ can be 2. It is known that the largest eigenvalue of $\Delta$ is 2 if and only if the graph has a non-trivial bipartite connected component (see, e.g., Chung & Graham (1997)). Therefore, we can develop a theory using the normalized Laplacian instead of the augmented one in parallel for such a graph $G$.

In Section 5, we characterized the asymptotic behavior of GCN defined by the augmented normalized Laplacian via its spectral distribution (Lemma 6 of Appendix D). We can derive a similar claim for GCN defined via the normalized Laplacian using the original theorem for the normalized Laplacian in Chung & Radcliffe (2011) (Theorem 7 therein). The normalized Laplacian version of GCN is advantegeous over the one made from the augmented one because we know its spectral distribution for broader range of random graphs. For example, Chung & Radcliffe (2011) proved the convergence of the spectral distribution of the normalized Laplacian for Chung-Lu's model (Chung & Lu, 2002), which includes power law graphs as a special case (see, Theorem 4 of Chung & Radcliffe (2011)).

## H   DETAILS OF EXPERIMENT SETTINGS

### H.1   EXPERIMENT OF SECTION 6.1

We set the eigenvalue of $P$ to $\lambda_1 = 0.5$ and $\lambda_2 = 1.0$ and randomly generated $P$ until the eigenvector $e$ associated to $\lambda_2$ satisfies the condition of each case described in the main article. We set $W = 1.2$ and used the following values for each case as $P$ and $e$.

#### H.1.1   CASE 1

$$P = \begin{bmatrix} 0.7469915 & 0.2499819 \\ 0.2499819 & 0.7530085 \end{bmatrix}, \quad e = \begin{bmatrix} 0.7028392 \\ -0.71134876 \end{bmatrix}.$$

#### H.1.2   CASE 2

$$P = \begin{bmatrix} 0.6899574 & -0.2426827 \\ -0.2426827 & 0.8100426 \end{bmatrix}, \quad e = \begin{bmatrix} 0.61637234 \\ -0.78745485 \end{bmatrix}.$$

### H.2   EXPERIMENT OF SECTION 6.2

We randomly generated an Erdős – Rényi graph $G_{N,p}$ with $N = 1000$ and randomly generated a one-of-$K$ hot vector for each node and embed it to a $C$-dimensional vector using a random matrix whose elements were randomly sampled from the standard Gaussian distribution. Here, $K = 10$ and $C = 32$. We treated the resulting single as the input signal $X^{(0)} \in \mathbb{R}^{N \times C}$. We constructed a GCN with $L = 10$ layers and $C$ channels. All parameters were i.i.d. sampled from the Gaussian distribution whose standard deviation is same as the one used in LeCun et al. (2012)[11] and multiplied a scalar to each weight matrix so that the largest singular value equals to a specified value $s$. We used three configurations $(p, s) = (0.1, 0.1), (0.5, 1.0), (0.5, 10.0)$. $\lambda$ of the generated GCNs are $0.063, 0.197, 0.194$, respectively. See Appendix 6.2 for the results of other configurations of $(p, s)$.

---

[11]This is the default initialization method for weight matrices in Chainer and Chainer Chemistry.

Table 1: Dataset specifications. The threshold $\lambda^{-1}$ in the table indicates the upper bound of Corollary 2.

|          | #Node | #Edge | #Class | Chance Rate | Threshold $\lambda^{-1}$ |
|----------|-------|-------|--------|-------------|--------------------------|
| Cora     | 2708  | 5429  | 6      | 30.2%       | $1 + 3.62 \times 10^{-3}$ |
| CiteSeer | 3312  | 4732  | 7      | 21.1%       | $1 + 1.25 \times 10^{-3}$ |
| PubMed   | 19717 | 44338 | 3      | 39.9%       | $1 + 9.57 \times 10^{-3}$ |

Table 2: Dataset specifications for noisy citation networks. The threshold $\lambda^{-1}$ in the table indicates the upper bound of Corollary 2.

|                 | Original Dataset | #Edge Added | Threshold $\lambda^{-1}$ |
|-----------------|------------------|-------------|--------------------------|
| Noisy Cora 2500 | Cora             | 2495        | 1.11                     |
| Noisy Cora 5000 | Cora             | 4988        | 1.15                     |
| Noisy CiteSeer  | CiteSeer         | 4991        | 1.13                     |
| Noisy PubMed    | PubMed           | 24993       | 1.17                     |

## H.3 EXPERIMENT OF SECTION 6.3

### H.3.1 DATASET

We used the Cora (McCallum et al., 2000; Sen et al., 2008), CiteSeer (Giles et al., 1998; Sen et al., 2008), and PubMed(Sen et al., 2008) datasets for experiments. We obtained the preprocessed dataset from the code repository of Kipf & Welling (2017)[12]. Table 1 summarizes specifications of datasets and their noisy version (explained in the next section).

The Cora dataset is a citation network dataset consisting of 2708 papers and 5429 links. Each paper is represented as the occurence of 1433 unique words and is associated to one of 7 genres (Case Based, Genetic Algorithms, Neural Networks, Probabilistic Methods, Reinforcement Learning, Rule Learning, Theory). The graph made from the citation links has 78 connected components and the smallest positive eigenvalue of the augmented Normalized Laplacian is approximately $\tilde{\mu} = 3.62 \times 10^{-3}$. Therefore, the upper bound of Theorem 2 is $\lambda^{-1} = (1 - \tilde{\mu})^{-1} \approx 1 + 3.62 \times 10^{-3}$. 818 out of 2708 samples are labelled as "Probabilistic Methods", which is the largest proportion. Therefore, the chance rate is $818/2708 = 30.2\%$.

The CiteSeer dataset is a citation network dataset consisting of 3312 papers and 4732 links. Each paper is represented as the occurence of 3703 unique words and is associated to one of 6 genres (Agents, AI, DB, IR, ML, HCI). The graph made from the citation links has 438 connected components and the smallest positive eigenvalue of the augmented Normalized Laplacian is approximately $\tilde{\mu} = 1.25 \times 10^{-3}$. Therefore, the upper bound of Theorem 2 is $\lambda^{-1} = (1 - \tilde{\mu})^{-1} \approx 1 + 1.25 \times 10^{-3}$. 701 out of 2708 samples are labelled as "IR", which is the largest proportion. Therefore, the chance rate is $701/3312 = 21.1\%$.

The PubMed dataset is a citation network dataset consisting of 19717 papers and 44338 links. Each paper is represented as the occurence of 500 unique words and is associated to one of 3 genres ("Diabetes Mellitus, Experimental", "Diabetes Mellitus Type 1", "Diabetes Mellitus Type 2"). The graph made from the citation links has 438 connected components and the smallest positive eigenvalue of the augmented Normalized Laplacian is approximately $\tilde{\mu} = 9.48 \times 10^{-3}$. Therefore, the upper bound of Theorem 2 is $\lambda^{-1} = (1 - \tilde{\mu})^{-1} \approx 1 + 9.57 \times 10^{-3}$. 7875 out of 19717 samples are labelled as "Diabetes Mellitus Type 2", which is the largest proportion. Therefore, the chance rate is $7875/19717 = 39.9\%$.

### H.3.2 NOISY CITATION NETWORKS

We summarize the properties of noisy citation networks in Table 2.

---

[12]https://github.com/tkipf/gcn

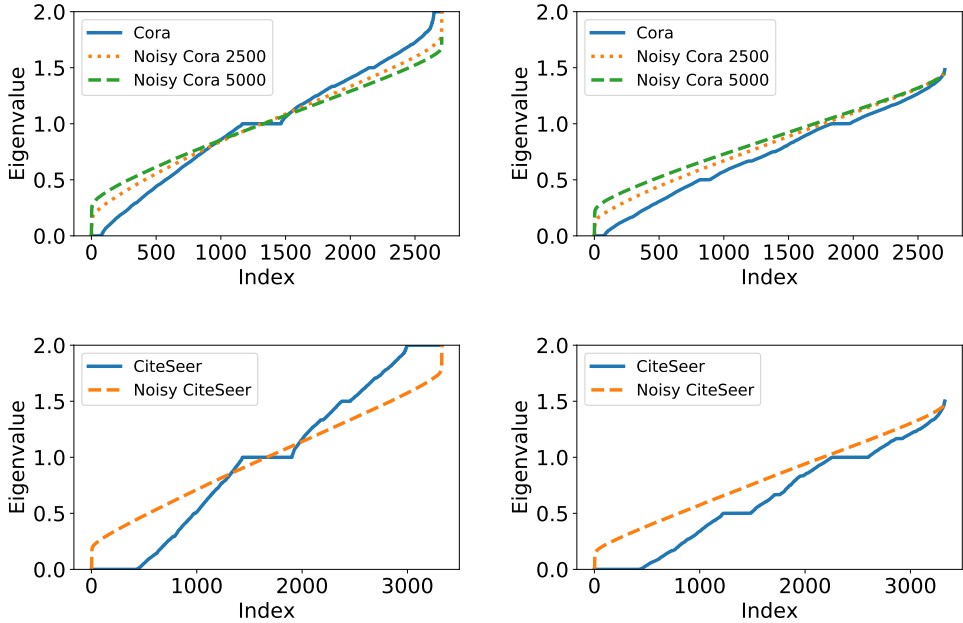

Figure 5: Spectral distribution of Laplacian for the citation network datasets. Left: normalized Laplacian. Right: augmented normalized Laplacian. Top: Cora and Noisy Cora (2500, 5000). Bottom: CiteSeer and Noisy CiteSeer.

We created two datasets from the Cora dataset: Noisy Cora 2500 and Noisy Cora 5000. *Noisy Cora 2500* is made from the Cora dataset by uniformly randomly adding 2500 edges, respectively. Since some random edges are overlapped with existing edges, the number of newly-added edges is 2495 in total. We only changed the underlying graph from the Cora dataset and did not change word occurences (feature vectors) and genres (labels). The underlying graph of the Noisy Cora dataset has two connected components and the smallest positive eigenvalue is $\tilde{\mu} \approx 9.62 \times 10^{-2}$. Therefore, the threshold of the maximum singular values of in Theorem 2 has been increased to $\lambda^{-1} = (1 - \tilde{\mu})^{-1} \approx 1.11$. Similarly, *Noisy Cora 5000* was made by adding 5000 edges uniformaly randomly. The number of newly added edges is 4988 and the graph is connected (i.e., it has only 1 connected component). $\tilde{\mu}$ and $\lambda$ are $\tilde{\mu} \approx 1.32 \times 10^{-1}$ and $\lambda = (1 - \tilde{\mu})^{-1} \approx 1.15$, respectively.

We made the noisy version of CiteSeer (*Noisy CiteSeer*) and PubMed (*Noisy PubMed*), in the similar way, by adding 5000 and 25000 edges uniformly randomly to the datasets. Since some random edges were overlapped with existing edges, 4991 and 24993 edges are newly added, respectively. This manipulation reduced the number of connected component of the graph to 3. $\tilde{\mu}$ is approximately $1.11 \times 10^{-1}$ (Noisy CiteSeer) and $1.43 \times 10^{-1}$ (Noisy PubMed) and $\lambda^{-1} = (1 - \tilde{\mu})^{-1}$ is approximately 1.13 (Noisy CiteSeer) and 1.17 (Noisy PubMed), respectively. Figure 5 (right) shows the spectral distribution of the augmented normalized Laplacian For comparison, we show in Figure 5 (left) the spectral distribution of the normalized Laplacian for these datasets[13].

### H.3.3 MODEL ARCHITECTURE

We used a GCN consisting of a single node embedding layer, one to nine graph convolution layers, and a readout operation (Gilmer et al., 2017), which is a linear transformation common to all nodes in our case. We applied softmax function to the output of GCN. The output dimension of GCN is same as the number of classes (i.e., seven for Noisy Cora 2500/5000, six for Noisy CiteSeer, and three for Noisy PubMed). We treated the number of units in each graph convolution layer as a hyperparameter. Optionally, we specified the maximum singular values $s$ of graph convolution

---

[13]Due to computational resource problems, we cannot compute the spectral distributions for PubMed and Noisy Pubmed.

Table 3: Hyperparameters of the experiment in Section 6.3. $X \sim \text{LogUnif}[10^a, 10^b]$ denotes the random variable $\log_{10} X$ obeys the uniform distribution over $[a, b]$. "Learning rate" corresponds to $\alpha$ when "Optimization algorithm" is Adam (Kingma & Ba, 2015).

| Name | Value |
|------|-------|
| Unit size | $\{10, 20, \ldots, 500\}$ |
| Epoch | $\{10, 20, \ldots, 100\}$ |
| Optimization algorithm | $\{\text{SGD}, \text{MomentumSGD}, \text{Adam}\}$ |
| Learning rate | $\text{LogUnif}[10^{-5}, 10^{-2}]$ |

layers. The choice of $s$ is either $0.5$ (smaller than 1), $s_1$ (in the interval $\{1 \leq s < \lambda^{-1}\}$), 3 and 10 (larger than $\lambda^{-1}$). We used $s_1 = 1.05$ for Noisy Cora 2500, Noisy CiteSeer, and Noisy PubMed, and $s_1 = 1.1$ for Noisy Cora 5000 and Noisy CiteSeer so that $s_1$ is not close to the edges of the the interval $\{1 \leq s < \lambda^{-1}\}$.

### H.3.4 PERFORMANCE EVALUATION PROCEDURE

We split all nodes in a graph (either Noisy Cora 2500/5000 or Noisy CiteSeer) into training, validation, and test sets. Data split is the same as the one done by Kipf & Welling (2017). This is a transductive learning (Pan & Yang, 2010) setting because we can use node properties of the validation and test data during training. We trained the model three times for each choice of hyperparemeters using the training set and defined the objective function as the average accuracy on the validation set. We chose the combination of hyperparameters that achieves the best value of objective function. We evaluate the accuracy of the test dataset three times using the chosen combination of hyperparameters and computed their average and the standard deviation.

### H.3.5 TRAINING

At initialization, we sampled parameters from the i.i.d. Gaussian distribution. If the scale of maximum singular values $s$ was specified, we subsequently scaled weight matrices of graph convolution layers so that their maximum singular values were normalized to $s$. The loss function was defined as the sum of the cross entropy loss for all training nodes. We train the model using the one of gradient-based optimization methods described in Table 3.

### H.3.6 HYPERPRAMETERS

Table 3 shows the set of hyperparameters from which we chose. Since we compute the representations of all nodes at once at each iteration, each epoch consists of 1 iteration. We employ Tree-structured Parzen Estimator (Bergstra et al., 2011) for hyperparameter optimization.

### H.3.7 IMPLEMENTATION

We used Chainer Chemistry[14], which is an extension library for the deep learning framework Chainer (Tokui et al., 2015; 2019), to implement GCNs and Optuna (Akiba et al., 2019) for hyperparameter tuning. We conducted experiments in a signel machine which has 2 Intel(R) Xeon(R) Gold 6136 CPU@3.00GHz (24 cores), 192 GB memory (DDR4), and 3 GPGPUs (NVIDIA Tesla V100). Our implementation achieved 68.1% with Dropout (Srivastava et al., 2014) (2 graph convolution layers) and 64.2% without Dropout (1 graph convolution layer) on the test dataset. These are slightly worse than the accuracy reported in Kipf & Welling (2017), but are still comparable with it.

### H.4 EXPERIMENT OF SECTION 6.4

The experiment settings are almost same as the experiment in Section 6.3. The only difference is that we did not train the node embedding layer, which we put before convolution layers of a GCN, while we did in Section 6.3. This is because we wanted to see the the effect of convolution operations

---

[14]https://github.com/pfnet-research/chainer-chemistry

on the perpendicular component of signals, while we interested in the prediction accuracy in real training settings in the previous experiment.

# I  ADDITIONAL EXPERIMENT RESULTS

## I.1  EXPERIMENT OF SECTION 6.1

We show the vector field $V$ for various $W$ in Figure 6 (**Case 1**) and Figure 7 (**Case 2**). Parameters other than $W$ are same as experiments in Section 6.1 (detail values are available in Appendix H.1).

## I.2  EXPERIMENT OF SECTION 6.2

Figure 8 shows the relative log distance of signals and their upper bound for various edge probability $p$ and the maximum singular value $s$. Note that we generate a new graph for each configuration of $(p, s)$. Therefore, different configurations may have different graphs and hence different $\lambda$ even they have a same edge probability $p$ in common.

## I.3  EXPERIMENT OF SECTION 6.3

### I.3.1  PREDICTIVE ACCURACY

Figure 9 shows the comparison of predictive performance in terms the maximum singular value and layer size when the dataset is Noisy Cora 5000 (left) and Noisy Citeseer (right), respectively. Concrete values are available in Table 4.

### I.3.2  TRANSITION OF MAXIMUM SINGULAR VALUES

Figure 10 – 13 show the transition of weight of graph convolution layers during training when the dataset is Noisy Cora 2500, Noisy Cora 5000, and Noisy CiteSeer, respectively. We note that the result of 3-layered GCN from the Noisy Cora 2500 is identical to Figure 3 (right) of the main article.

## I.4  EXPERIMENT OF SECTION 6.4

Figure 14 shows the logarithm of relative perpendicular component and prediction accuracy on Noisy Cora, Noisy CiteSeer, and Noisy PubMed datasets. We use Pearson R as a correlation coefficient. If GCNs have only one layer, it has more large relative perpendicular components (corresponding to right points in the figures) than GCNs which have other number of layers. The correlation between the logarithm of relative perpendicular components and prediction accuracies are $0.827(p = 6.890 \times 10^{-6})$ for Noisy Cora, $0.524(p = 1.771 \times 10^{-2})$ for Noisy CiteSeer, and $0.679(p = 1.002 \times 10^{-3})$ for Noisy PubMed, if we treat the one-layer case as outliers and remove them.

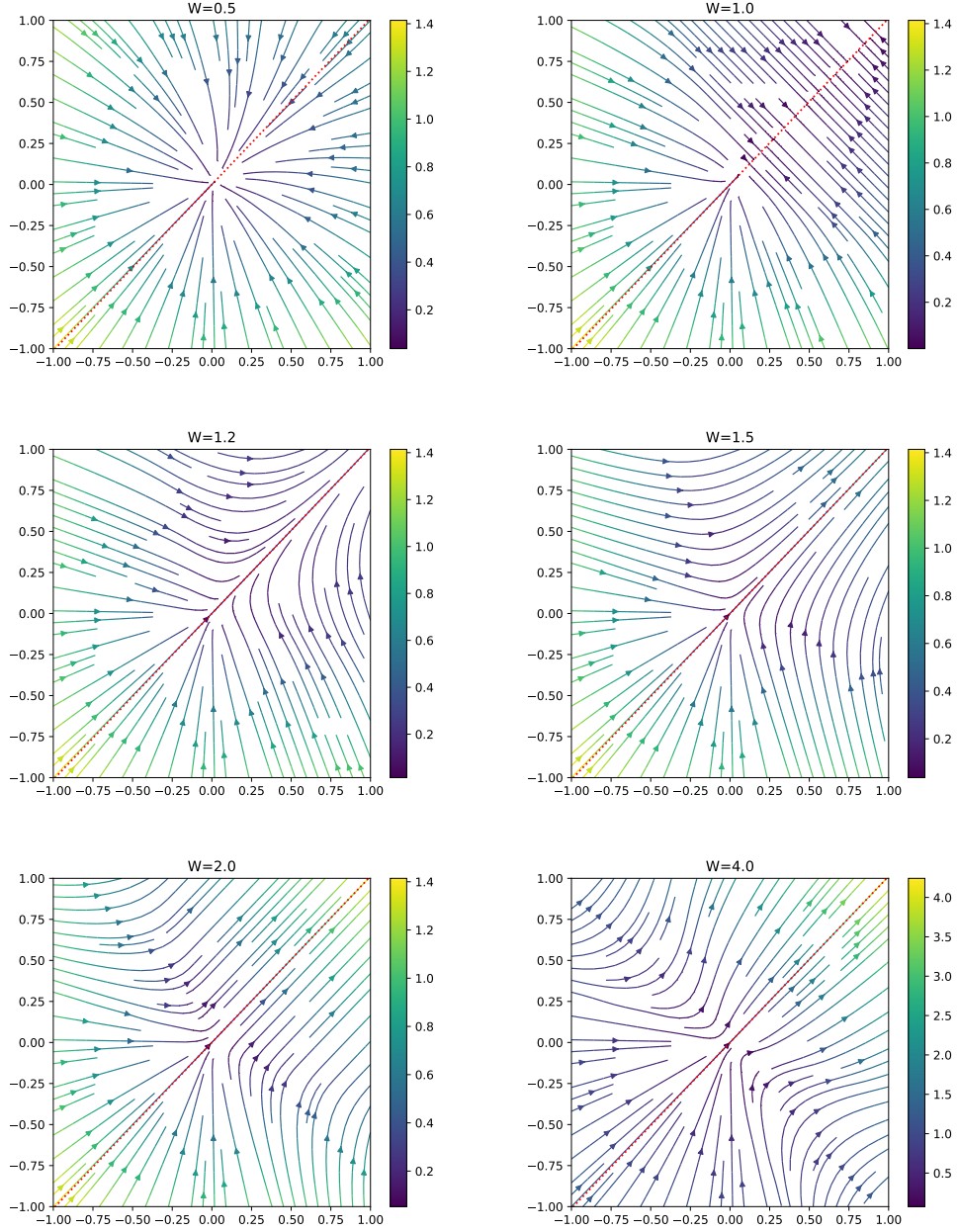

Figure 6: Vector field $V$ for various $W$ for **Case 1**. Top left: $W = 0.5$. Top right: $W = 1.0$. Middle left: $W = 1.2$ (same as Figure 1 in the main article). Middle right: $W = 1.5$. Bottom left: $W = 2.0$. Bottom right: $W = 4.0$. Best view in color.

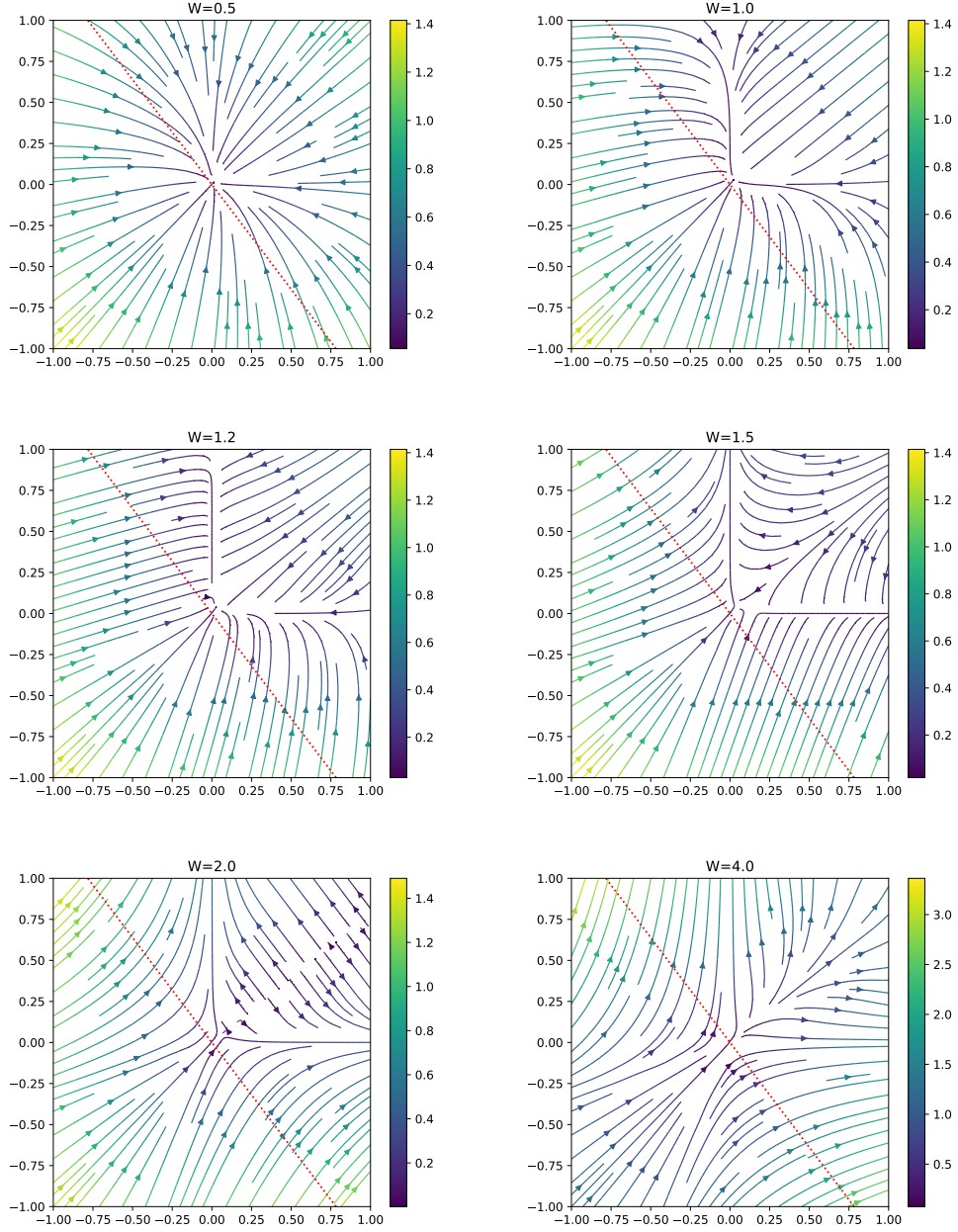

Figure 7: Vector field $V$ for various $W$ for **Case 2**. Top left: $W = 0.5$. Top right: $W = 1.0$. Middle left: $W = 1.2$ (same as Figure 1 in the main article). Middle right: $W = 1.5$. Bottom left: $W = 2.0$. Bottom right: $W = 4.0$. Best view in color.

Table 4: Comparison of performance in terms of maximum singular value of weights and layer size. "U" in the right most column indicates the accuracy of GCN without weight normalization.

### Noisy Cora 2500

| | Maximum Singular Value | | | | |
|---|---|---|---|---|---|
| Depth | 1 | 1.05 | 3 | 10 | U |
| 1 | $0.389 \pm 0.101$ | $0.429 \pm 0.090$ | $0.552 \pm 0.014$ | $0.632 \pm 0.007$ | $0.587 \pm 0.008$ |
| 3 | $0.273 \pm 0.051$ | $0.309 \pm 0.017$ | $0.580 \pm 0.058$ | $0.661 \pm 0.003$ | $0.494 \pm 0.041$ |
| 5 | $0.319 \pm 0.000$ | $0.267 \pm 0.059$ | $0.462 \pm 0.065$ | $0.602 \pm 0.004$ | $0.326 \pm 0.029$ |
| 7 | $0.261 \pm 0.076$ | $0.262 \pm 0.080$ | $0.407 \pm 0.021$ | $0.501 \pm 0.017$ | $0.279 \pm 0.129$ |
| 9 | $0.261 \pm 0.080$ | $0.319 \pm 0.000$ | $0.284 \pm 0.109$ | $0.443 \pm 0.014$ | $0.319 \pm 0.000$ |

### Noisy Cora 5000

| | Maximum Singular Value | | | | |
|---|---|---|---|---|---|
| Depth | 1 | 1.1 | 3 | 10 | U |
| 1 | $0.301 \pm 0.080$ | $0.333 \pm 0.099$ | $0.557 \pm 0.004$ | $0.561 \pm 0.019$ | $0.555 \pm 0.016$ |
| 3 | $0.245 \pm 0.066$ | $0.247 \pm 0.076$ | $0.370 \pm 0.041$ | $0.587 \pm 0.009$ | $0.286 \pm 0.066$ |
| 5 | $0.274 \pm 0.048$ | $0.237 \pm 0.070$ | $0.257 \pm 0.076$ | $0.535 \pm 0.031$ | $0.319 \pm 0.000$ |
| 7 | $0.263 \pm 0.080$ | $0.297 \pm 0.031$ | $0.260 \pm 0.074$ | $0.339 \pm 0.060$ | $0.319 \pm 0.000$ |
| 9 | $0.262 \pm 0.081$ | $0.258 \pm 0.064$ | $0.262 \pm 0.080$ | $0.261 \pm 0.082$ | $0.318 \pm 0.002$ |

### Noisy CiteSeer

| | Maximum Singular Value | | | | |
|---|---|---|---|---|---|
| Depth | 0.5 | 1.1 | 3 | 10 | U |
| 1 | $0.461 \pm 0.018$ | $0.467 \pm 0.012$ | $0.490 \pm 0.016$ | $0.494 \pm 0.006$ | $0.495 \pm 0.009$ |
| 3 | $0.438 \pm 0.027$ | $0.436 \pm 0.010$ | $0.450 \pm 0.019$ | $0.462 \pm 0.007$ | $0.417 \pm 0.061$ |
| 5 | $0.285 \pm 0.008$ | $0.371 \pm 0.016$ | $0.373 \pm 0.011$ | $0.425 \pm 0.007$ | $0.380 \pm 0.024$ |
| 7 | $0.213 \pm 0.006$ | $0.282 \pm 0.011$ | $0.309 \pm 0.012$ | $0.385 \pm 0.007$ | $0.308 \pm 0.012$ |
| 9 | $0.182 \pm 0.005$ | $0.242 \pm 0.030$ | $0.303 \pm 0.021$ | $0.325 \pm 0.003$ | $0.229 \pm 0.033$ |

### Noisy Pubmed

| | Maximum Singular Value | | | | |
|---|---|---|---|---|---|
| Depth | 0.5 | 1.1 | 3 | 10 | U |
| 1 | $0.488 \pm 0.039$ | $0.636 \pm 0.006$ | $0.641 \pm 0.010$ | $0.632 \pm 0.002$ | $0.631 \pm 0.010$ |
| 3 | $0.442 \pm 0.027$ | $0.426 \pm 0.026$ | $0.658 \pm 0.004$ | $0.661 \pm 0.005$ | $0.631 \pm 0.013$ |
| 5 | $0.431 \pm 0.033$ | $0.431 \pm 0.034$ | $0.561 \pm 0.083$ | $0.641 \pm 0.004$ | $0.424 \pm 0.093$ |
| 7 | $0.428 \pm 0.032$ | $0.443 \pm 0.051$ | $0.449 \pm 0.035$ | $0.619 \pm 0.011$ | $0.440 \pm 0.041$ |
| 9 | $0.413 \pm 0.009$ | $0.438 \pm 0.039$ | $0.539 \pm 0.052$ | $0.569 \pm 0.042$ | $0.473 \pm 0.031$ |

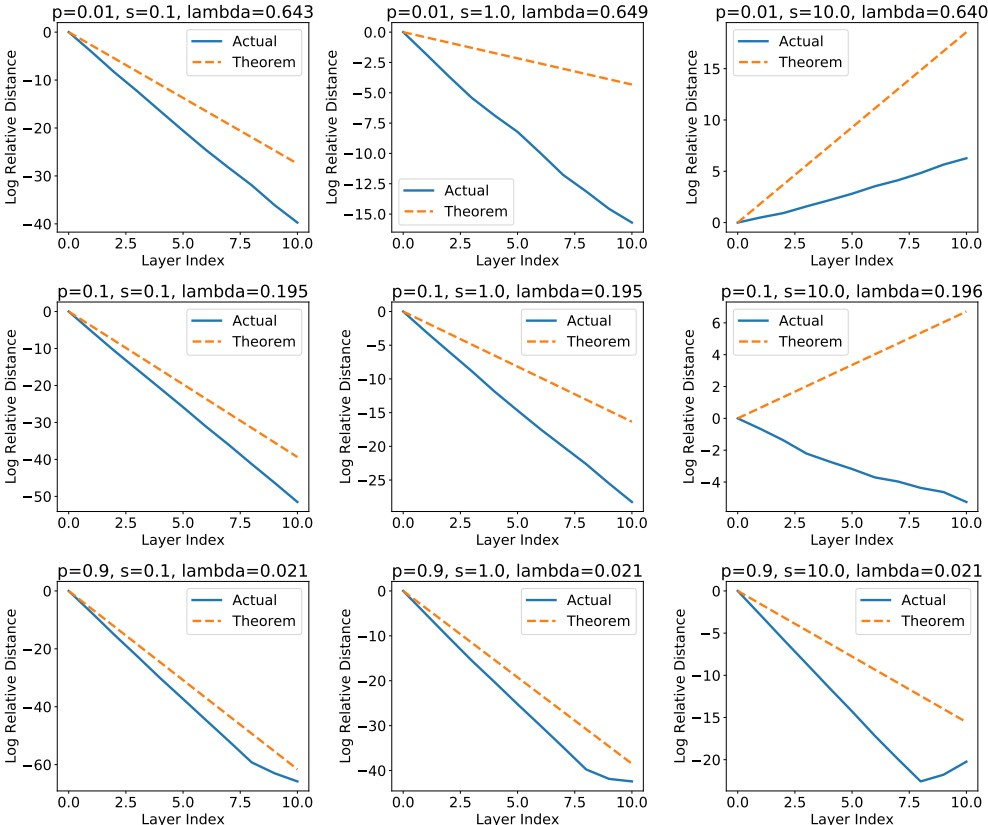

Figure 8: The actual distance $d_\mathcal{M}$ to the invariant space $\mathcal{M}$ and the upper bound inferred by Theorem 1. The edge probability $p$ takes $0.01(\mathrm{top}), 0.1, 0.9(\mathrm{bottom})$ and the maximum singular value $s$ takes $0.1(\mathrm{left}), 1.0, 10(\mathrm{right})$. Blue lines are the log relative distance defined by $y(l) := \log \frac{d_\mathcal{M}(X^{(l)})}{d_\mathcal{M}(X^{(0)})}$ and orange dotted lines are upper bound $y(l) := l \log(s\lambda)$, where $X^{(0)}$ is the input signal and $X^{(l)}$ is the output of the $l$-th layer. Best view in color.

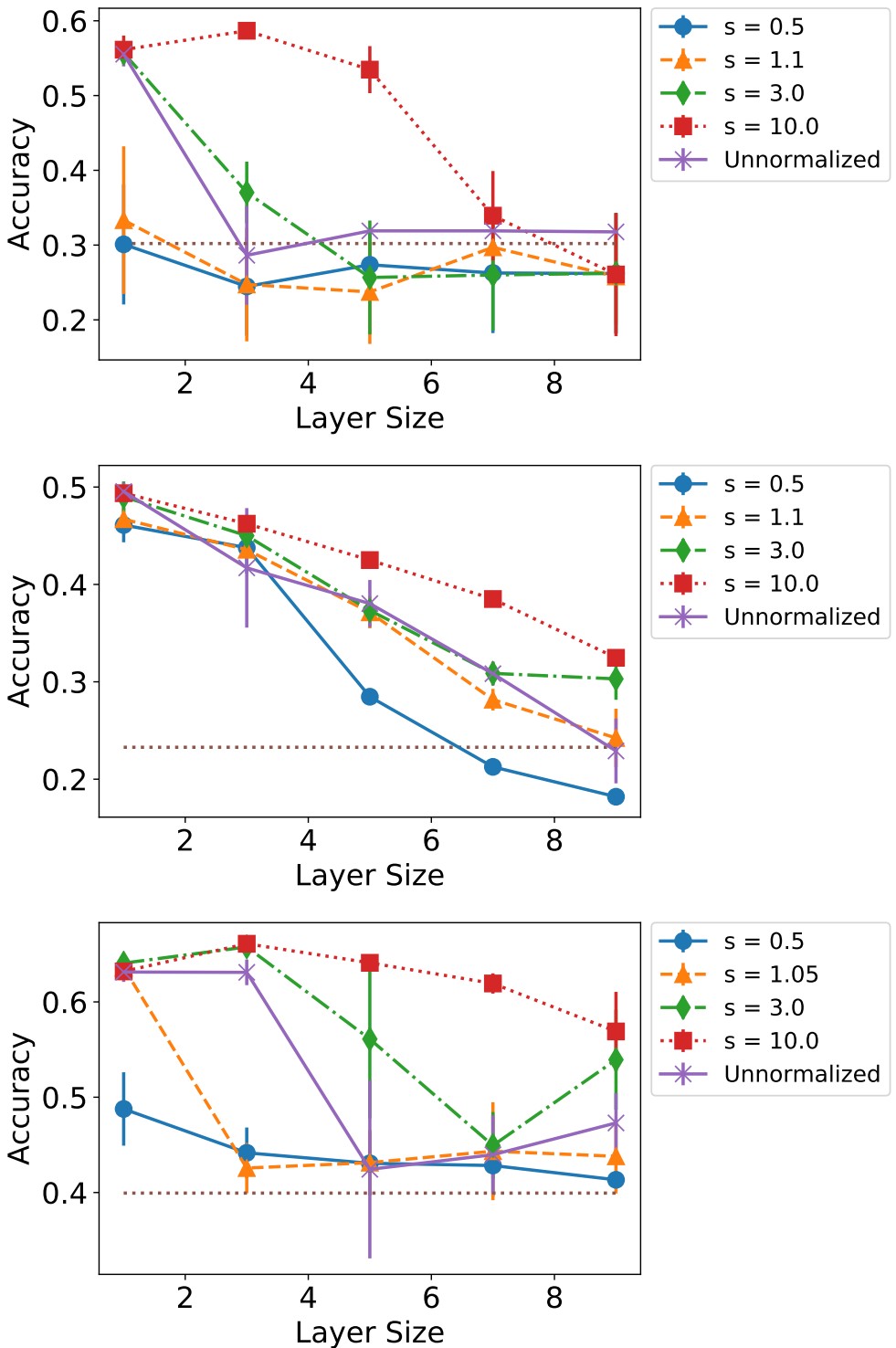

Figure 9: Effect of the maximum singular values of weights on predictive performance. Horizontal dotted lines indicate the chance rates (30.2% for Noisy Cora 5000, 21.2% for Noisy CiteSeer, and 39.9% for Noisy PubMed). The error bar is the standard deviation of 3 trials. Left: Noisy Cora 5000. Right: Noisy CiteSeer. Bottom: Noisy Pubmed. Best view in color.

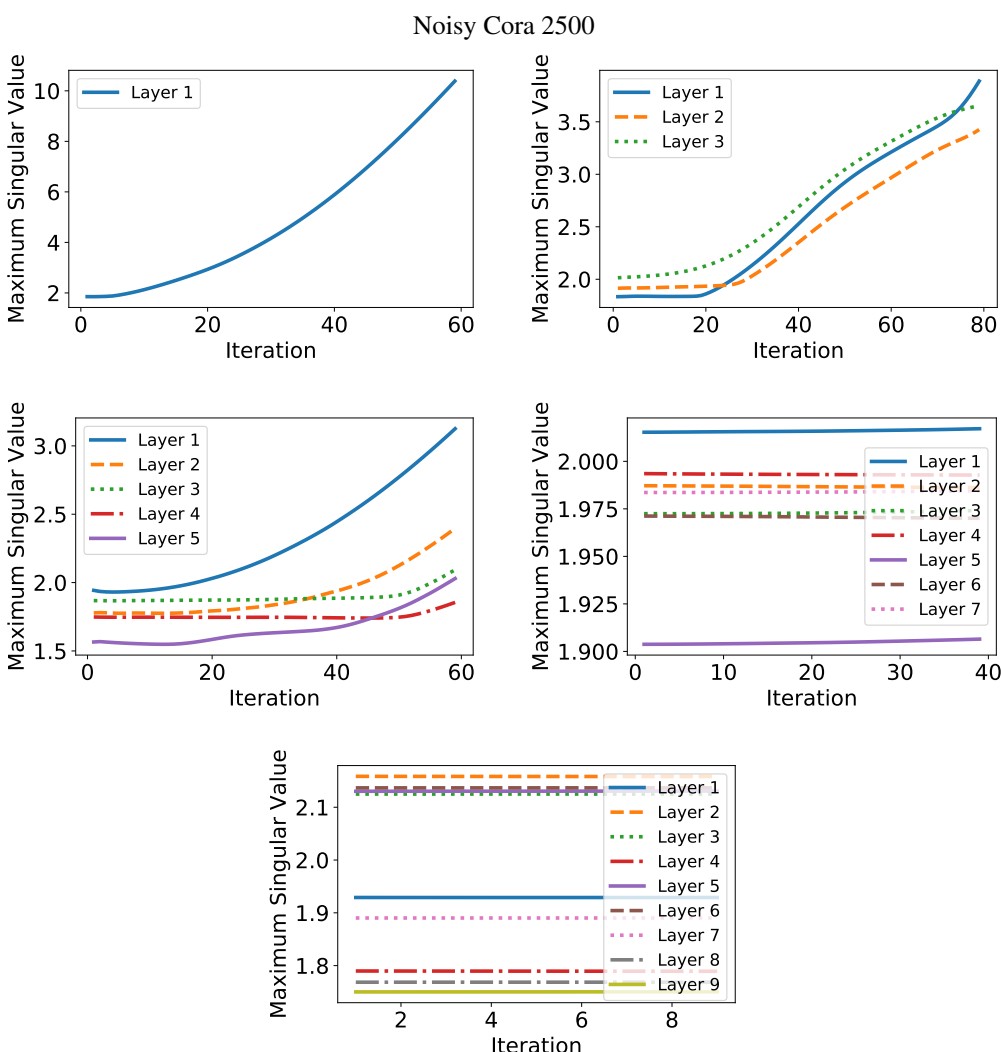

Figure 10: Transition of maximum singular values of GCN during training using Noisy Cora 2500. Top left: 1 layer. Top right: 5 layers. Bottom left: 7 layers. Bottom right: 9 layers.

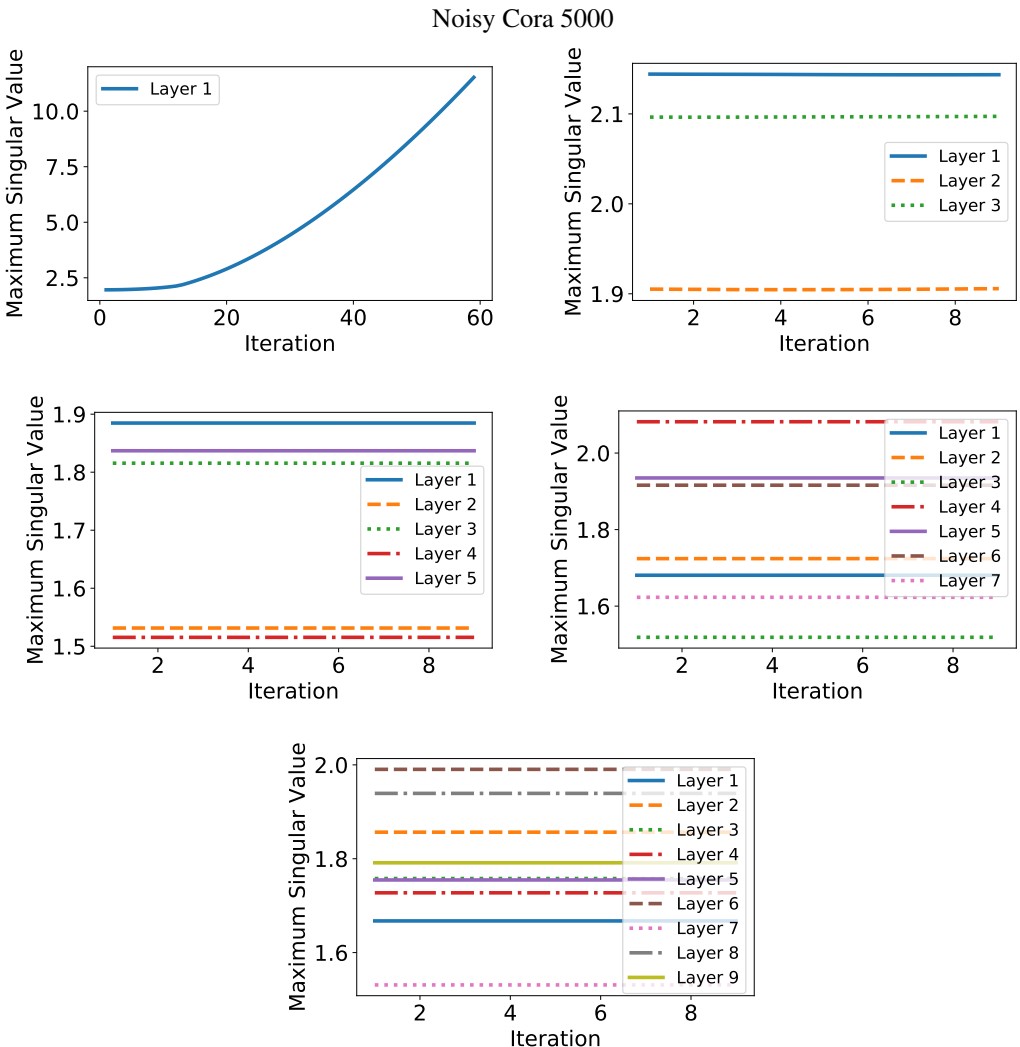

Figure 11: Transition of maximum singular values of GCN during training using Noisy Cora 5000. Top left: 1 layer. Top right: 3 layers. Middle left: 5 layers. Middle right: 7 layers. Bottom: 9 layers.

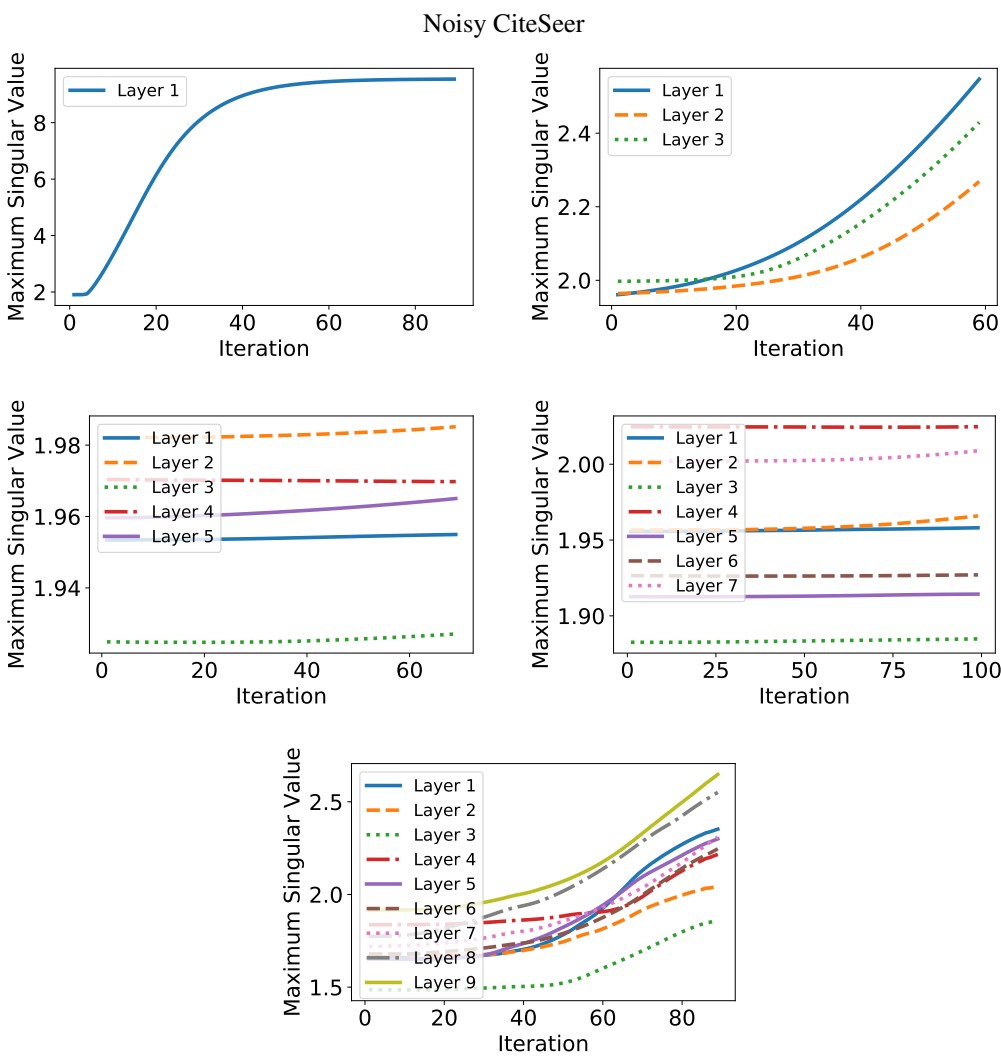

Figure 12: Transition of maximum singular values of GCN during training using Noisy CiteSeer. Top left: 1 layer. Top right: 3 layers. Middle left: 5 layers. Middle right: 7 layers. Bottom: 9 layers.

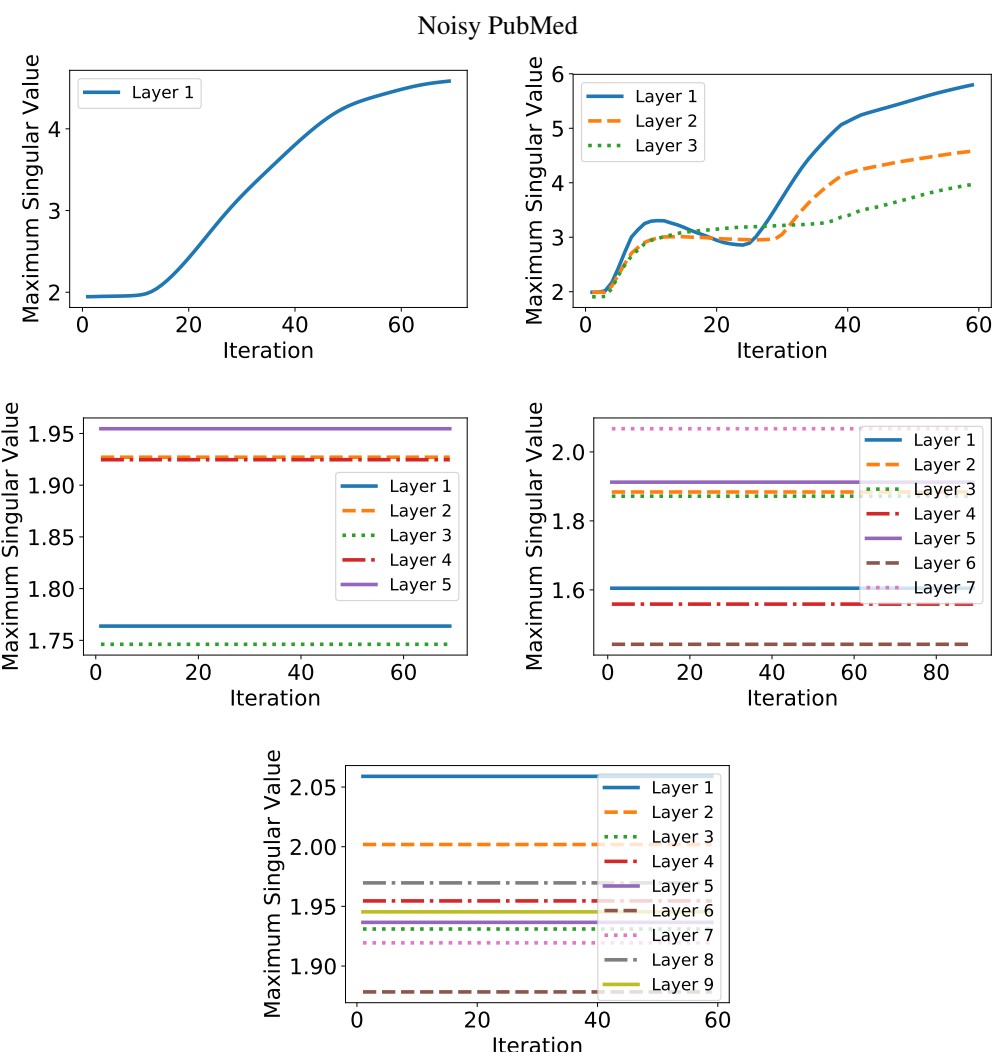

Figure 13: Transition of maximum singular values of GCN during training using Noisy PubMed. Top left: 1 layer. Top right: 3 layers. Middle left: 5 layers. Middle right: 7 layers. Bottom: 9 layers.

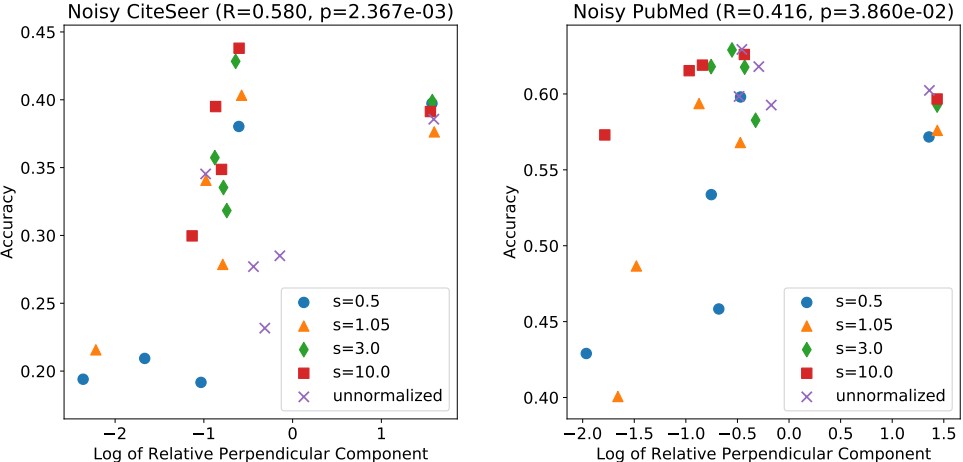

Figure 14: Logarithm of relative perpendicular component and prediction accuracy. Left: Noisy CiteSeer. Right: Noisy PubMed. $p$ in the title represents the $p$-value for the Pearson R coefficients.

