# OpenReview forum: "Graph Neural Networks Exponentially Lose Expressive Power for Node Classification"
_ICLR.cc/2020/Conference — Accept (Spotlight)_

### Official Review · AnonReviewer3 · 2019-10-20
**Official Blind Review #3**

**Rating:** 8

**Review:**

In the paper, the authors carry out theoretically analysis on the expressive power for GNN. The analysis focused on the limiting case when the depth of layers goes to infinite. The authors prove that if the weights of the GNN satisfy certain condition based on the graph Laplacian, then the transformed features contain only degree and connected component information. Then the authors study the G_{np} random graph as a special case. Finally, empirical experiments are carried out to corroborates the theoretical results.

Strength:
1. The authors study a very important problem. It has long been observed that additional depth does not help GNN. The authors provide a convincing theoretical explanation for the behavior.
2. The theoretical results are very close to practical models with little assumptions. Most importantly, the non-linearity is kept compared to other analysis that takes essential components out from the model. Moreover, the technique used in the paper could potentially benefit other theoretical analysis for NN.
3. The authors provide supporting empirical experiments for the theoretical analysis. The observation on s in section 6.3 and the perpendicular component experiment on section 6.4 provide interesting insight to the performance of GNN.

Weakness:
1. The analysis is mostly for dense graphs. However, most of the real-world networks are large-scale sparse networks.
2. It would be great if the authors could provide suggestions on how to eliminate the information loss in GNN. For example, it would be super interesting if the authors can generalize their analysis when residual links exists.


**Experience Assessment:**

I have read many papers in this area.

**Review Assessment: Checking Correctness Of Derivations And Theory:**

I assessed the sensibility of the derivations and theory.

**Review Assessment: Checking Correctness Of Experiments:**

I assessed the sensibility of the experiments.

**Review Assessment: Thoroughness In Paper Reading:**

I read the paper at least twice and used my best judgement in assessing the paper.

---

> ### Author Response · Authors · 2019-11-14
> **Response to Reviewer #3**
>
> First of all, thank you for your valuable feedback and comments. We appreciate that you highly evaluate our work.
>
>
> > The analysis is mostly for dense graphs. However, most of the real-world networks are large-scale sparse networks.
>
> First, we think that Erdos-Renyi graphs Theorem 3 can support are not that dense. For example, suppose the node size is $N = 20000$ (approximate node size of the PubMed dataset) and the edge probability $p = \log N / N$, each node has the order of $Np=\log N\approx 4.3$ adjacent nodes (we use the common logarithm).
> In addition, although Theorem 3 cannot support the extremely sparse case (e.g., $p=o(\log N / N$) for Erdos-Renyi graphs), our theory gives a unified way to investigate the behavior of graph NNs regardless of the sparsity of underlying graphs through graph spectral.
> Regarding experiments, we can obtain insights about sparse graphs that the sparsity of practically available graphs can be one of the reasons for the success of graph NNs for node classification tasks. To confirm this hypothesis, we artificially added edges to citation networks to make them dense in the experiments and observed the failure of graph NNs as expected. Of course, our experiments are just supporting evidence of the hypothesis. We need more theories to understand the oversmoothing phenomena for sparse graphs.
>
>
> > It would be great if the authors could provide suggestions on how to eliminate the information loss in GNN.
>
> One idea is to (randomly) sample edges in an underlying graph. As we wrote in the previous answer, the sparsity of practically available graphs could be a factor of the success of graph NNs. Assuming this hypothesis is correct, there is a possibility that we can relive the effect of oversmoothing by sparsification. Since we can never restore the information in pruned edges if we remove them permanently, random edge sampling could work better as FastGCN [Chen et al., 2018] or GraphSAGE [Hamilton et al., 2017] do.
> Another idea is to scale a signal (i.e., intermediate or final output of graph NNs) appropriately so that the signal keeps away from the invariant space $\mathcal{M}$. Our proposed weight scaling mechanism takes this strategy. Recently, Zhao and Akoglu (2019) have proposed PairNorm to alleviate the oversmoothing phenomena. Although the scaling target is different -- they rescaled signals whereas ours normalized weights -- there can be some relationship between theirs and ours.
>
>
> [Chen et al., 2018] Jie Chen,  Tengfei  Ma,  and Cao Xiao.  FastGCN: Fast learning with  graph convolutional networks via importance sampling. InInternational Conference on Learning Representations, 2018.
> [Hamilton et al., 2017] Will Hamilton, Zhitao Ying, and Jure Leskovec.  Inductive representation learning on large graphs.   In Advances  in  Neural  Information  Processing  Systems  30,  pages  1024–1034, 2017.
> [Zhao and Akoglu, 2019]
>
>
> > For example, it would be super interesting if the authors can generalize their analysis when residual links exists.
>
> We thank the reviewer for suggestions. We are also interested in how residual links affect behaviors of graph NNs (both in the finite-depth regime and in the continuous-limit regime). Considering the correspondence of graph NNs and Markov processes (see Remark 1 and Appendix E), we thought at first that residual links do not contribute to alleviating the oversmoothing phenomena: our intuition was that adding residual connections to a graph NN corresponds to converting a Markov process to the lazy version. When a Markov process converges to a stable distribution, the corresponding lazy process also converges eventually. It implies that residual links might not be helpful.
> However, Li et al. (2019) reported that graph NNs with as many as 56 layers performed well if they added residual connections. Considering that, the situation could be more complicated than our intuitions. We think it is a promising direction for future research.
>
> [Li et al., 2019] Guohao Li, Matthias Mller, Ali Thabet, and Bernard Ghanem. DeepGCNs: Can GCNs go as deep as CNNs? In The IEEE International Conference on Computer Vision (ICCV), 2019.

---

### Official Review · AnonReviewer2 · 2019-10-23
**Official Blind Review #2**

**Rating:** 6

**Review:**

This paper proposes that the outputs for distinguish nodes exponentially approach signals that only carry topological information of the graph, when the weights in GCN satisfy conditions determined by spectra of the augmented normalized Laplacian, by generalizing the forward propagation of GCN to a specific dynamical system. With the guidance of this theory, it experimentally confirm weight scaling enhances the predictive performance of GCNs in both synthesized and real data.

Overall, this paper could be a considerable theoretical contribution. I recommend a weak accept for this work. It explains the observation that GCNs do not improve (or sometimes worsen) the performance as we pile up more layers, which is indeed we met in our work. It also points out a useful technique, weights normalization, in training a GCN. As mentioned in the paper, scaling the weights is a trade-off between information loss and generalization error, which is an interesting topic worthing further exploration.

However, there are one question needing more clarification. Although the output are being projected into the null space of the Laplacian, it does mean the signals only carry topological information. Distinguish nodes can still have different outputs that allows good classification results.

**Experience Assessment:**

I have published one or two papers in this area.

**Review Assessment: Checking Correctness Of Derivations And Theory:**

I assessed the sensibility of the derivations and theory.

**Review Assessment: Checking Correctness Of Experiments:**

I carefully checked the experiments.

**Review Assessment: Thoroughness In Paper Reading:**

I read the paper at least twice and used my best judgement in assessing the paper.

---

> ### Author Response · Authors · 2019-11-13
> **Response to Reviewer #2**
>
> Thank you for your feedback and comments. We appreciate that you highly evaluate our work.
>
> > Although the output are being projected into the null space of the Laplacian, it does mean the signals only carry topological information. Distinguish nodes can still have different outputs that allows good classification results.
>
> First, we would like to confirm that what the reviewer wanted to say was that "it does NOT mean the signals only carry topological information" and "Different nodes can have different outputs." If so, we would say that our claim is different from the reviewer's understanding. As we wrote in the paragraph right after Theorem 2, the null space of the Laplacian does only carry topological information in the sense that if two nodes have the same node degrees and are in the same connected component, the corresponding output is identical. We appreciate it if the reviewer lets us know if we misunderstand the intention.

---

### Official Review · AnonReviewer1 · 2019-10-26
**Official Blind Review #1**

**Rating:** 8

**Review:**

The paper studies why graph NNs lose the expressive power as additional layers are added. A dynamical system perspective is adopted and used to show that under certain conditions on the weights, the expressiveness of the network deteriorates. This is since the network's output eventually only carries information about superficial graph properties for distinguishing nodes, and nothing else. A case study of Erdos-Renyi graph is provided, showing that for dense graphs information loss indeed occurs. Guidelines to try and deal with this in practice were devised and empirically examined.

I enjoyed reading the paper, and found the results interesting and informative. The presentation is good and the derivations are clear. I do not have significant criticism, but would like address a few points (see below). In sum, the paper should be accepted in my opinion.

Comments:

1) Understanding the behavior in sparse graphs seems like an important research avenue, without which the picture is incomplete. Perhaps adding some empirical results in that direction?

2) The method for transforming sparse graphs into dense ones in Section 6.3 is not convincing. The random addition of edges compromises the truth correlation profile of citations. What should we learn about the actual task from the experiments on the noisy version?

3) In relation to generalization error, mentioned in the discussion section. Could the double descent curves of https://arxiv.org/abs/1812.11118 reconcile the discussion therein? How would over-parameterization from the interpolation perspective and the results of the current work relate?

**Experience Assessment:**

I do not know much about this area.

**Review Assessment: Checking Correctness Of Derivations And Theory:**

I assessed the sensibility of the derivations and theory.

**Review Assessment: Checking Correctness Of Experiments:**

I assessed the sensibility of the experiments.

**Review Assessment: Thoroughness In Paper Reading:**

I read the paper at least twice and used my best judgement in assessing the paper.

---

> ### Author Response · Authors · 2019-11-14
> **Response to Reviewer #1**
>
> We thank for your positive feedback and comments. We reply to your comments one by one.
>
>
> > Understanding the behavior in sparse graphs seems like an important research avenue, without which the picture is incomplete.
>
> First, we think that Erdos-Renyi graphs Theorem 3 can support are not that dense. For example, suppose the node size is $N=20000$ (approximate node size of the PubMed dataset) and the edge probability $p=\log N / N$, each node has the order of $Np=\log N\approx 4.3$ adjacent nodes (we use the common logarithm).
> In addition, although Theorem 3 cannot support the extremely sparse case (e.g., $p=o(\log N / N$) for Erdos-Renyi graphs), our theory gives a unified way to investigate the behavior of graph NNs regardless of the sparsity of underlying graphs through graph spectral.
> Regarding experiments, see the answer to the next question.
>
>
> > What should we learn about the actual task from the experiments on the noisy version?
>
> Our theory claims that GCNs tend to fail when graphs are dense (e.g., Theorem 3).  From this, we can hypothesize that the sparsity of practically available graphs is one of the reasons for the success of graph NNs for node classification tasks. To confirm this hypothesis, we artificially added edges to citation networks to make them dense in the experiments and observed the failure of graph NNs as expected. Of course, our experiments are just supporting evidence of the hypothesis. We need more theories to understand the oversmoothing phenomena for sparse graphs.
>
>
> > Could the double descent curves [...] reconcile the discussion therein? How would over-parameterization from the interpolation perspective and the results of the current work relate?
>
> To the best of our knowledge, no literature reported the double descent phenomena for graph NNs. Therefore, we had in mind the classical U-shaped risk curve, in which the bias-variance trade-off exists, and conjectured that there is a sweet spot of the weight scale (see the discussion section for detail). We are happy if some people share research that has studied it.
> It is known that the double descent phenomena do not occur in some situations, especially depending on regularization types. For example, while Belkin et al. (2018), the first paper of the double descent, employed the interpolating hypothesis with the minimum norm, Mei and Montanari (2019) found that the double descent was alleviated or disappeared when they used Ridge-type regularizations. Therefore, one can hypothesize the oversmoothing is a cause or consequence of regularization that is more like a Ridge-type rather than minimum-norm inductive bias.
>
> [Belkin et al., 2018] Belkin, Mikhail, et al. "Reconciling modern machine learning and the bias-variance trade-off." arXiv preprint arXiv:1812.11118 (2018).
> [Mei and Montanari, 2019] Mei, Song, and Andrea Montanari. "The generalization error of random features regression: Precise asymptotics and double descent curve." arXiv preprint arXiv:1908.05355 (2019).

---

### Public Comment · ~Keyulu_Xu1 · 2019-10-05
**Excellent theory and related work**

This is an amazing piece of theoretical work and we enjoyed reading it.

We (ERATO Kawarabayashi Large Graph Project, at NII, Japan, https://bigdata.nii.ac.jp/wp/contact/) would like to draw your attention that we have related work showing similar but non-asymptotic properties of GCN from the lens of expander graph, tree-width and random walk theory. We hope to have further discussions and references with you.

Representation Learning on Graphs with Jumping Knowledge Networks. K. Xu, C. Li, Y. Tian, T. Sonobe, K. Kawarabayashi, S. Jegelka. ICML 2018.

---

> ### Author Response · Authors · 2019-10-08
> **Thank you for your interest.**
>
> Thank you for taking an interest in our paper, and sorry for our late reply. I am happy to have further discussions with you.
>
> I like your work, which has elegantly related graph NNs with random walks. Since the random walk is closely related to graph spectra via, e.g., heat equations, I think your analysis is related to ours for explaining empirical phenomena of graph NNs such as oversmoothing.
>
> The interesting point is how to overcome the difficulty coming from the non-linearity. If I understand correctly, you have assumed the randomized activation function and justified the assumption empirically, whereas we used the fact that ReLU is a projection onto a cone {X\geq 0\}$.
>
> We considered Erdos-Renyi graphs as a prototypical case. We may say more precise behaviors of graph NNs if we specialize graphs on which we define graph NNs as you did.

---

> > ### Public Comment · ~Keyulu_Xu1 · 2019-10-10
> > **Re: Thank you for your interest.**
> >
> > Thank you for liking our work. It is very interesting analysis for ReLU using the the fact that ReLU is a projection onto a cone $\{X\geq 0\}$. Great work! Using the assumptions from Kawaguchi 16' is essentially linearizing the network, though we found that it aligns with practice well for node classification tasks.
> >
> > From a more general perspective, i.e. beyond the node classification datasets, we find such "graph expander phenomenon" is subtle for more general tasks. For example, in reasoning tasks, e.g. VQA, learning graph algorithms, people usually use complete graphs, which for sure have high expansion, but  more layers is usually needed and not harmful. So this has to do with the "true underlying function" of these node classification datasets (maybe that's also why adaptivity of aggregation range for these node classification tasks is desirable). This is an observation we had after JK-Nets so unfortunately we did not have chance to include this in final version of JK, but we think this would reveal a better picture of what's going on. Would be great if you add the references and further discussions in your final version.

---

### Author Response · Authors · 2020-02-12
**Camera Ready Uploaded**

We thank reviewers, area chair, and Keyulu Xu, who engaged in the discussion, for valuable feedback and comments. And we thank all people who read our paper. We have uploaded the camera ready version of our paper.

Updates
- Added several references to Related work, including JK-net (Xu et al., 2018), and papers about the over-smoothing for linear GNNs.
- Added more remarks on Theorem 3, based on review comments.
- Reflected comments from reviewers to Discussion.
- Added comments on the relationship between Luan et al. (2019), which addressed the over-smoothing of non-linear graph neural networks, and ours (Remark 2, Appendix C).
- Uploaded the experiment code to Github: https://github.com/delta2323/gnn-asymptotics
- Improved wording and images. Fixed typos.

We welcome any feedback or comments.

Kenta Oono and Taiji Suzuki

---

### Author Response · Authors · 2020-10-20
**New version uploaded**

We found several mistakes in Proposition 3. As guided by the organizer, we have updated our paper, along with some other improvements. Specifially, we applied the following changes:

- Proposition 3: we fix the statement (no less than $\lambda$ → no larger than $\lambda$).
- Corollary 3: we remove a redundant assumption ($\lambda < |\lambda_N|$).
- Remark 2 and Section C: Since Luan et al. (2019) upload the camera-ready version, we explicitly write that the counterexample we made is for the theorem in the old preprint (version 2).
- We update acknowledgement.
- Other minor improvements.

Best
Kenta Oono

---

### Decision · Program_Chairs · 2019-12-19

**Decision:**

Accept (Spotlight)

**Comment:**

The paper provides a theoretical analysis of graph neural networks, as the number of layers goes to infinity. For the graph convolutional network, they relate the expressive power of the network with the graph spectra. In particular for Erdos-Renyi graphs, they show that very deep graphs lose information, and propose a new weight normalization scheme based on this insight.

The authors responded well to reviewer comments. It is nice to see that the open review nature has also resulted in a new connection. Unfortunately one of the reviewers did not engage further in the discussion with respect to the author rebuttals.

Overall, the paper provides a nice theoretical analysis of a widely used graph neural network architecture, and characterises its behaviour on a popular class of graphs. The fact that the theory provides a new approach for weight normalization is a bonus.